# Tunable and low-loss correlated plasmons in Mott-like insulating oxides

Teguh Citra Asmara[1,2,*], Dongyang Wan[1,3], Yongliang Zhao[1,*], Muhammad Aziz Majidi[2,3,*], Christopher T. Nelson[4,5], Mary C. Scott[4,5], Yao Cai[4], Bixing Yan[1,3], Daniel Schmidt[2], Ming Yang[2,3], Tao Zhu[2,3], Paolo E. Trevisanutto[2,3,6], Mallikarjuna R. Motapothula[1,3], Yuan Ping Feng[3,6], Mark B.H. Breese[1,2,3], Matthew Sherburne[4], Mark Asta[4], Andrew Minor[4,5], T. Venkatesan[1,3,7,8,9] & Andrivo Rusydi[1,2,3,7]

Plasmonics has attracted tremendous interests for its ability to confine light into subwavelength dimensions, creating novel devices with unprecedented functionalities. New plasmonic materials are actively being searched, especially those with tunable plasmons and low loss in the visible–ultraviolet range. Such plasmons commonly occur in metals, but many metals have high plasmonic loss in the optical range, a main issue in current plasmonic research. Here, we discover an anomalous form of tunable correlated plasmons in a Mott-like insulating oxide from the $Sr_{1-x}Nb_{1-y}O_{3+\delta}$ family. These correlated plasmons have multiple plasmon frequencies and low loss in the visible–ultraviolet range. Supported by theoretical calculations, these plasmons arise from the nanometre-spaced confinement of extra oxygen planes that enhances the unscreened Coulomb interactions among charges. The correlated plasmons are tunable: they diminish as extra oxygen plane density or film thickness decreases. Our results open a path for plasmonics research in previously untapped insulating and strongly-correlated materials.

[1] NUSNNI-NanoCore, National University of Singapore, Singapore 117411, Singapore. [2] Singapore Synchrotron Light Source, National University of Singapore, Singapore 117603, Singapore. [3] Department of Physics, National University of Singapore, Singapore 117551, Singapore. [4] Department of Materials Science and Engineering, University of California, Berkeley, California 94720, USA. [5] National Center for Electron Microscopy, Molecular Foundry, Lawrence Berkeley National Laboratory, Berkeley, California 94720, USA. [6] Centre for Advanced 2D Materials and Graphene Research Centre, National University of Singapore, Singapore 117546, Singapore. [7] NUS Graduate School for Integrative Sciences and Engineering, National University of Singapore, Singapore 117456, Singapore. [8] Department of Electrical and Computer Engineering, National University of Singapore, Singapore 117583, Singapore. [9] Department of Materials Science and Engineering, National University of Singapore, Singapore 117575, Singapore. * These authors contributed equally to this work. Correspondence and requests for materials should be addressed to T.V. (email: venky@nus.edu.sg) or to A.R. (email: phyandri@nus.edu.sg).

Plasmonics offers a crossroad between photonics and nanoelectronics by combining the former's high-bandwidth capability with the latter's nanoscale integrability[1–5]. Plasmonics utilizes plasmon, a collective excitation of charges that arises from interactions between electromagnetic fields (such as photons) and free charges[3]. Conventionally, the plasmon frequency depends on the free-charge density[3]; thus, high-frequency plasmons are usually found in metals[3–5] due to their abundance of free charges, but are rarely observed in conventional wide bandgap insulators.

In strongly-correlated materials, conventional forms of plasmons have been observed in the metallic or superconducting phases of these materials, and there have been studies to explore their potential for plasmonics in vanadium oxides[6–8] and cuprates[9,10]. In particular, localized conventional surface plasmons have been observed in the metallic phase of $VO_2$, and its temperature-dependent metal–insulator transition has been utilized for plasmonic switching and sensing[6–8]. However, in strongly-correlated, insulating phases of these materials (such as in Mott insulators[11]), correlated forms of plasmons under long-range Coulomb interactions have only been theoretically investigated[12] but not experimentally observed.

Meanwhile, the $Sr_{1−x}Nb_{1−y}O_{3+\delta}$ family of oxides are known to have rich structures and electronic properties that vary with oxygen content. For example, perovskite $SrNbO_3$ is conducting with an Nb-$4d^1$ electronic structure[13–16], and $Sr_{1−x}NbO_3$ (ref. 14) has been found to potentially be a good photocatalyst in water splitting applications[15]. On the other hand, oxygen-rich $SrNbO_{3.4}$ and $SrNbO_{3.5}$, derived by interspersing the perovskite lattice of $SrNbO_3$ with extra oxygen planes along the [101] direction at certain periodic intervals[16,17] (see Supplementary Note 1 for details), is a quasi-one-dimensional conductor[16–21] and a ferroelectric insulator[22,23], respectively. Previously, there are no reported studies on plasmons in this family of oxides.

In this study, several $Sr_{1−x}NbO_{3+\delta}$ films with varying oxygen content, electrical conductivity (from metallic to insulator-like) and film thicknesses are studied using spectroscopic ellipsometry (SE), atomic-resolution transmission electron microscopy (TEM), transport measurements and supported by theoretical calculations based on coupled harmonic oscillators model and density functional theory. The results show a surprising observation of a new form of correlated plasmons in the insulator-like film, itself also revealed to be a strongly-correlated Mott-like insulator. The correlated plasmons unusually have multiple plasmon frequencies ($\sim 1.7$, $\sim 3.0$ and $\sim 4.0$ eV) and low loss (several times lower than gold) in the visible–ultraviolet range. Supported by theoretical calculations, these correlated plasmons arise from collective excitations of correlated electrons in the film, where the nanometre-spaced confinement of extra oxygen planes causes increased Coulomb repulsions among the electrons. The correlated plasmons are reproducible and tunable: they diminish and ultimately vanish as extra oxygen plane density or film thickness decreases. In particular, the decrease of extra oxygen plane density increases the electrical conductivity, and, as correlated plasmons are vanishing in the metallic films, the increased free-charge density causes conventional plasmon to arise at $\sim 1.9$ eV.

## Results

**Transport measurements results**. The main batch of pressure-dependent $Sr_{1−x}NbO_{3+\delta}$ films is deposited on (001)LaAlO$_3$ substrates using pulsed-laser deposition under three oxygen pressure conditions: $5 \times 10^{−6}$, $3 \times 10^{−5}$ and $1 \times 10^{−4}$ Torr, labelled as lp-SNO, mp-SNO and hp-SNO, respectively. Transport measurements reveal that lp-SNO and mp-SNO are

metallic with room-temperature resistivity of $1 \times 10^{−4}$ and $6 \times 10^{−3}\,\Omega$ cm, respectively, while hp-SNO is insulator-like with room-temperature resistivity of $6\,\Omega$ cm. The conducting lp-SNO have an unusually large room-temperature free-charge density of $\sim 1 \times 10^{22}$ cm$^{−3}$, while for mp-SNO it is $\sim 4 \times 10^{21}$ cm$^{−3}$. The room-temperature mobility of the conducting films is $\sim 2.5$ cm$^2$ V$^{−1}$ s$^{−1}$ for lp-SNO and $\sim 0.3$ cm$^2$ V$^{−1}$ s$^{−1}$ for mp-SNO. The thickness of the three main films is $\sim 196$, $\sim 218$ and $\sim 168$ nm for lp-SNO, mp-SNO and hp-SNO, respectively. For thickness-dependent study, thinner hp-SNO films with varying thicknesses of $\sim 81$, $\sim 52$ and $\sim 20$ nm are also deposited. For reproducibility, a second batch of pressure-dependent films is deposited as well under six oxygen pressure conditions: $5 \times 10^{−6}$ Torr (lp-SNO-2), $1 \times 10^{−5}$ Torr (mlp-SNO-2), $3 \times 10^{−5}$ Torr (mp-SNO-2), $7 \times 10^{−5}$ Torr (mhp-SNO-2), $1 \times 10^{−4}$ Torr (hp-SNO-2) and $5 \times 10^{−4}$ (hrp-SNO-2). The lp-SNO-2, mlp-SNO-2, mp-SNO-2, and mhp-SNO-2 films are conducting, while the hp-SNO-2 and hrp-SNO-2 films are insulating, and their thicknesses are kept to be within a narrow range of $\sim 159$–$182$ nm (see Supplementary Table 1 for details). The crystal structures of the representative main batch films are studied using X-ray diffractions (XRDs) shown in Supplementary Fig. 1.

**Complex dielectric function and loss function**. Figure 1 shows the complex dielectric function, $\varepsilon(\omega) = \varepsilon_1(\omega) + i\varepsilon_2(\omega)$, and loss function (LF), $-\mathrm{Im}[\varepsilon^{−1}(\omega)] = \frac{\varepsilon_2(\omega)}{\varepsilon_1^2(\omega) + \varepsilon_2^2(\omega)}$, of the pressure-dependent $Sr_{1−x}NbO_{3+\delta}$ films extracted from SE, where $\omega$ is the photon angular frequency (see Supplementary Figs 2–5 for details). Each peak in the transverse $\varepsilon(\omega)$ spectra indicates an optical excitation, while each peak in the longitudinal LF spectra indicates a plasmonic excitation. For sub-X-ray photons, the photon momentum transfer, $\mathbf{q}$, is finite but approaches zero because it is much less than the crystal momentum. In this limit, the distinction between longitudinal (l) and transverse (t) $\varepsilon(\omega)$ vanishes, that is, $\lim_{|\mathbf{q}|\to 0} \varepsilon_l(\mathbf{q},\omega) = \varepsilon_t(\mathbf{q},\omega)$, which allows sub-X-ray optical spectroscopy to probe both optical and plasmonic properties of materials in the low-$\mathbf{q}$ limit[24]. All of the optical and plasmonic peaks are listed in Table 1. It should be noted that longitudinal LF spectrum is also the quantity probed by electron energy loss spectroscopy, which is another common technique used in the study of plasmonic properties of materials. However, as hp-SNO is insulator-like, such photoemission-based spectroscopy measurements might be challenging for hp-SNO due to significant charging effects.

The $\varepsilon_2(\omega)$ of lp-SNO (Fig. 1b) shows an intraband Drude peak ($A_1$) and the first interband transition peak ($A_2$ and $A_3$, $> 4.6$ eV), which has been attributed as O-$2p \to$ Nb-$4d$ transition[15,16]. (Note that in the cited studies, the theoretical bandgap between O-$2p$ and Nb-$4d$ calculated using density functional theory was underestimated[25]. Nevertheless, the $A_2$ peak from $\sim 4.6$ eV should still be regarded as originated from O-$2p \to$ Nb-$4d$ transition because it is the lowest-energy interband transition in lp-SNO.) The LF of lp-SNO (Fig. 1b) shows a large peak ($A'_1$, $\sim 1.9$ eV), which coincides with the zero-crossing of $\varepsilon_1(\omega)$ at $\sim 1.84$ eV (Fig. 1a) and is very close to the reflectivity minimum at $\sim 2.1$ eV (Supplementary Fig. 6), indicating that the peak comes from a conventional plasmon excitation[3]. There is a slight blueshift between the $A'_1$ peak and the $\varepsilon_1(\omega)$ zero-crossing due to the free-electron scattering. Consequently, from this blueshift along with the full-width half-maximum (FWHM) of the $A'_1$ peak (see equations 5 and 6), we can estimate the free-electron scattering, $1/\tau$, in lp-SNO to be $\sim 0.47$ eV and its conventional plasmon dephasing time[3], $T_2$, to be $\sim 2.8$ fs.

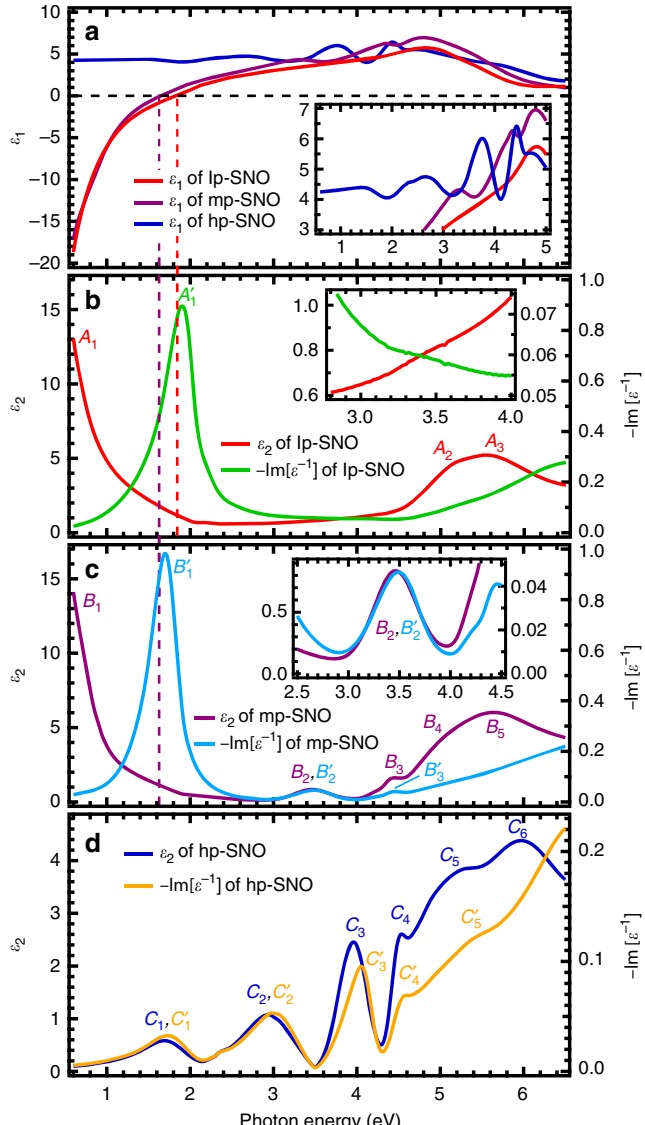

**Figure 1 | Complex dielectric function and loss function spectra of pressure-dependent Sr$_{1-x}$NbO$_{3+\delta}$ films.** (**a**) Real part of complex dielectric function, $\varepsilon_1(\omega)$, of lp-SNO ($5 \times 10^{-6}$ Torr), mp-SNO ($3 \times 10^{-5}$ Torr) and hp-SNO ($1 \times 10^{-4}$ Torr) films. Vertical dashed lines indicate the zero-crossings of $\varepsilon_1(\omega)$ of lp-SNO (red) and mp-SNO (purple) films. (**b**) Imaginary part of complex dielectric function, $\varepsilon_2(\omega)$, and loss function, $-\text{Im}[\varepsilon^{-1}(\omega)]$, spectra of lp-SNO film. (**c**) The $\varepsilon_2(\omega)$ and $-\text{Im}[\varepsilon^{-1}(\omega)]$ spectra of mp-SNO film. (**d**) The $\varepsilon_2(\omega)$ and $-\text{Im}[\varepsilon^{-1}(\omega)]$ spectra of hp-SNO film. Insets show parts of the spectra zoomed in for clarity.

Meanwhile, in the less-conducting mp-SNO, the conventional plasmon peak ($B'_1$ in Fig. 1c) redshifts to $\sim 1.7$ eV due to the decrease in free-charge density. From the FWHM of the $B'_1$ peak and its blueshift with respect to the $\varepsilon_1(\omega)$ zero-crossing at $\sim 1.62$ eV, the $1/\tau$ and $T_2$ in mp-SNO can be estimated to be $\sim 0.52$ eV and $\sim 2.5$ fs, respectively. The reflectivity minimum also redshifts to $\sim 1.9$ eV (Supplementary Fig. 6). Besides the Drude ($B_1$) and the first interband transition peaks ($B_4$ and $B_5$), the $\varepsilon_2(\omega)$ of mp-SNO also shows two additional peaks at $\sim 3.5$ ($B_2$) and $\sim 4.5$ eV ($B_3$). Particularly for $B_2$, it shares the same shape and energy position with the $B'_2$ of the corresponding LF spectrum. As discussed below, $B'_2$ is a new form of correlated plasmon, with a different origin from the $B'_1$ conventional plasmon.

An important observation is shown in the $\varepsilon_2(\omega)$ and LF spectra of hp-SNO (Fig. 1d). The $\varepsilon_2(\omega)$ shows that hp-SNO has no apparent Drude peak, consistent with transport measurement, with a wide bandgap of $\sim 4.6$ eV ($C_5$ and $C_6$). Interestingly, below this bandgap there exist several mid-gap peaks at $\sim 1.7$ ($C_1$), $\sim 3.0$ ($C_2$), $\sim 4.0$ ($C_3$) and $\sim 4.5$ eV ($C_4$) with low overall $\varepsilon_2(\omega)$ ($< 1$ for $C_1$ and $C_2$). Particularly, $C_1$, $C_2$ and $C_3$ peaks share very similar shapes and energy positions with another group of three peaks in the LF spectrum: $C'_1$, $C'_2$ and $C'_3$, respectively (just like the similarities between $B_2$ and $B'_2$ peaks of mp-SNO). The energy positions of both group of peaks are also very close to the reflectivity minima of hp-SNO (Supplementary Fig. 6) at $\sim 1.9$, $\sim 3.2$ and $\sim 4.2$ eV. These indicate that the two groups of peaks come from a similar origin. The appearance of these mid-gap peaks in the LF spectrum indicates that they arise from plasmonic excitations[3] and not from other effects such as spin–orbit coupling, phononic, excitonic or anisotropic effects (see Supplementary Note 2). Meanwhile, their concurrent existence in $\varepsilon_2(\omega)$ spectrum indicates that there is a coupling between optical and plasmonic excitations in hp-SNO (and to a lesser extent, in mp-SNO). As discussed later, these mid-gap plasmonic excitations are born from collective excitations of correlated electrons, and are thus called correlated plasmons.

As a first approximation, the dephasing time of these correlated plasmons may be estimated from FWHM of each peak. The FWHM of $C'_1$, $C'_2$, and $C'_3$ peaks is $\sim 0.7$, $\sim 0.68$ and $\sim 0.38$ eV, respectively. This leads to the dephasing time of $\sim 1.9$, $\sim 1.9$ and $\sim 3.5$ fs, respectively. We note that the scattering and dephasing mechanisms in correlated systems might be different from those in weakly correlated metals due to long-range correlation effects, as discussed later.

Meanwhile, the $C_4$ peak may be excitonic in origin due to its relative sharpness, asymmetric shape and energy position that is just below the bandgap. This peak diminishes as free-charge density increases in conducting mp-SNO (see the similar $B_3$ peak in mp-SNO), and ultimately vanishes in lp-SNO. If this peak is indeed excitonic, its presence in hp-SNO indicates that electron–hole interactions are unscreened in the insulating hp-SNO film, while its diminishing behaviour in mp-SNO and lp-SNO indicates that electron–hole interactions are heavily screened in the conducting films.

**Differences between correlated and conventional plasmons.** The correlated plasmons of hp-SNO are fundamentally different from conventional (bulk) plasmon in lp-SNO, conventional metals such as gold[3,5,26] and the metallic phases of strongly correlated materials such as metallic $VO_2$ (refs 6–8) for the following reasons. First, since hp-SNO is insulator-like, the correlated plasmons do not originate from collective excitations of free charges. In fact, as free-charge density increases in mp-SNO, the correlated plasmons instead become weaker and ultimately vanish in lp-SNO. Second, in lp-SNO and gold the $\varepsilon_1(\omega)$ is negative below the conventional plasmon energy and conventional plasmons occur at the zero-crossing of $\varepsilon_1(\omega)$ (albeit with slight blueshifts due to the free-electron scattering), while the $\varepsilon_1(\omega)$ of hp-SNO stays positive below the correlated plasmon energies. Third, conventional plasmons as in lp-SNO and gold usually only have one (bulk) plasmon energy, while hp-SNO has at least three observable correlated plasmon energies with a seemingly ordered energy ratio. This is somewhat similar to a previous theoretical result[12] where the presence of long-range Coulomb interaction in correlated systems could induce multiple correlated plasmon energies to appear. These theoretical plasmon energies also had an ordered ratio of $U^*$ and $U^*/2$, where $U^*$ was the effective local Coulomb interaction, although the energy ratio

**Table 1 | Optical and plasmonic excitations in pressure-dependent $Sr_{1-x}NbO_{3+\delta}$ films.**

| Optical ($\varepsilon(\omega)$) | | Plasmonic (loss function) | |
|---|---|---|---|
| **Peak** | **Energy (eV)** | **Peak** | **Energy (eV)** |
| *lp-SNO* | | | |
| $A_1$ | 0 | $A'_1$ | ~1.9 |
| $A_2$ | ~4.6–~5.3 | — | — |
| $A_3$ | ~5.3–~6.5 | — | — |
| | | | |
| *mp-SNO* | | | |
| $B_1$ | 0 | $B'_1$ | ~1.7 |
| $B_2$ | ~3.5 | $B'_2$ | ~3.5 |
| $B_3$ | ~4.5 | $B'_3$ | ~4.5 |
| $B_4$ | ~4.6–~5.2 | — | — |
| $B_5$ | ~5.2–~6.5 | — | — |
| | | | |
| *hp-SNO* | | | |
| $C_1$ | ~1.7 | $C'_1$ | ~1.7 |
| $C_2$ | ~3.0 | $C'_2$ | ~3.0 |
| $C_3$ | ~4.0 | $C'_3$ | ~4.0 |
| $C_4$ | ~4.5 | $C'_4$ | ~4.6 |
| $C_5$ | ~4.6–~5.5 | $C'_5$ | ~5.5 |
| $C_6$ | ~5.5–~6.5 | — | — |

The peaks in the imaginary part of complex dielectric function, $\varepsilon_2(\omega)$, and loss function, $-\mathrm{Im}\,[\varepsilon^{-1}(\omega)]$, spectra of lp-SNO ($5 \times 10^{-6}$ Torr), and hp-SNO ($1 \times 10^{-4}$ Torr) films are listed along with their photon energy positions. The peaks at 0 eV in lp-SNO and mp-SNO are Drude peaks associated with metallic free charges.

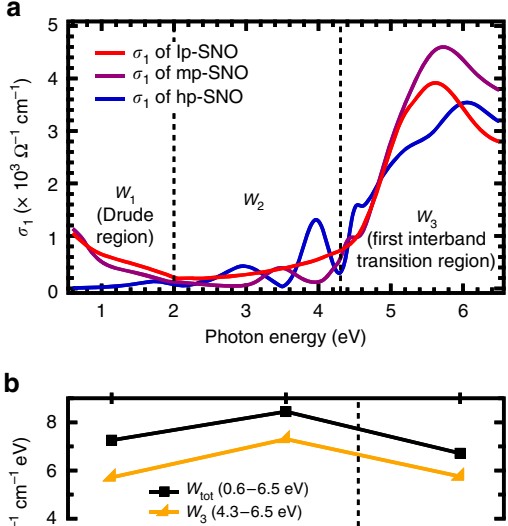

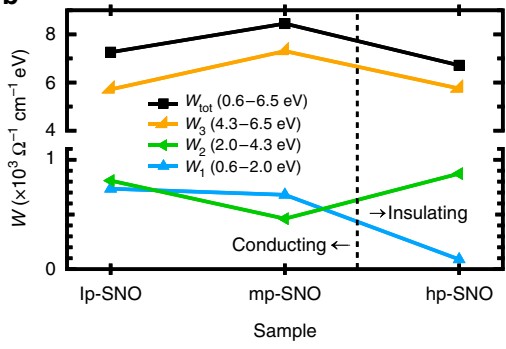

**Figure 2 | Optical conductivity spectra and spectral weight of pressure-dependent $Sr_{1-x}NbO_{3+\delta}$ films.** (**a**) The optical conductivity, $\sigma_1(\omega)$, of lp-SNO ($5 \times 10^{-6}$ Torr), mp-SNO ($3 \times 10^{-5}$ Torr) and hp-SNO ($1 \times 10^{-4}$ Torr) films. (**b**) Evolution of spectral weight, $W$, of three energy regions: $W_1$ (0.6–2.0 eV), $W_2$ (2.0–4.3 eV) and $W_3$ (4.3–6.5 eV) across the three films. The $W_{tot}$ is the total $W$ from 0.6 to 6.5 eV.

of the observed correlated plasmons of hp-SNO is not as straightforward to deduce (see Supplementary Note 3 for details). Fourth, the correlated plasmons appear in both $\varepsilon_2(\omega)$ and LF spectra of hp-SNO, which means that the correlated plasmons can readily be excited by, decay radiatively into and thus couple with free-space photons without any external mechanisms. This is unlike conventional surface plasmon resonance in metals, which needs external phase-matching mechanisms such as grating, Otto and Kretschmann configurations[3,27,28] to couple the plasmons with free-space photons. Intriguingly, the $\varepsilon_2(\omega)$ of hp-SNO below its bandgap is several times (for example, ~6 times at 3.0 eV) lower than that of gold[3,5,26], which means the correlated plasmons have low loss in the optical visible–ultraviolet range. To explain these differences, we further study the optical conductivity ($\sigma_1(\omega) = \varepsilon_0\varepsilon_2(\omega)\omega$, where $\varepsilon_0$ is the vacuum permittivity), spectral weight transfer and atomic structure of the films.

**Optical conductivity and spectral weight analysis.** The $\sigma_1(\omega)$ analysis (Fig. 2a) is important because it obeys the $f$-sum (charge conservation) rule[24,29,30], $\int_0^\infty \sigma_1(\omega)d\omega = \frac{\pi n e^2}{2m_e}$, where $n$, $e$ and $m_e$ is the total electron density, elementary charge and electron mass, respectively. From this rule, a partial spectral weight integral of an energy region (for example, from $E_1$ to $E_2$) can be defined as $W = \int_{E_1}^{E_2} \sigma_1(E)dE$. The $W$ is proportional to the effective number of electrons participating in the optical excitations, which means by analysing the evolution of $W$ we can study the various charge transfers that occur in the films and gauge their electronic correlations[11,24,30–35]. For this purpose, the $\sigma_1(\omega)$ spectra are divided into three energy regions: $W_1$ for the Drude peak (0.6–2.0 eV), $W_2$ for the mid-gap plasmonic peaks (2.0–4.3 eV) and $W_3$ for the first interband transition of O-2$p$→Nb-4$d$ (4.3–6.5 eV). The evolution of each $W$ across the three films is shown in Fig. 2b (see also Supplementary Fig. 7 for more details).

As free-charge density decreases from lp-SNO to mp-SNO, the decrease of $W_1$ (Drude region) is accompanied by an increase of $W_3$. This can be understood because the decreased number of

electrons in the conduction band (Nb-4$d$) increases the available unoccupied states necessary for the first interband transition of O-2$p$→Nb-4$d$. Thus, as free-charge density further decreases and ultimately vanishes during a transition from metal to insulator, the diminishing of Drude peak is expected to be accompanied by a further increase of $W_3$. Surprisingly, Fig. 2 shows this is not the case for the metal–insulator transition (MIT) between mp-SNO and hp-SNO, because instead $W_1$ and $W_3$ both anomalously decrease as MIT occurs, leading to an overall decrease of $W$ below 6.5 eV ($W_{tot}$). According to the $f$-sum rule[24,29,30] and because hp-SNO is insulator-like with no apparent Drude peak, this decrease has to be compensated by an equivalent increase of $W$ above 6.5 eV, implying spectral weight transfers over wide energy ranges on the onset of the MIT. Such wide-range anomalous spectral weight transfers are a direct evidence of strong electronic correlation[11,24,30–35] and has also been observed in other strongly-correlated materials such as cuprates[11,30,33], vanadium oxides[34] and manganites[11,30,35]. This anomalous spectral weight transfer behaviour signifies that hp-SNO is most likely a Mott-like insulator and the MIT occurs as the Coulomb repulsion between electrons becomes stronger (that is, unscreened) and electronic correlation increases. This also means that the plasmons in hp-SNO are a new type of correlated plasmons born from collective excitations of correlated electrons.

**Transmission electron microscopy.** The three films are also studied using TEM to determine their microstructures. Atomic resolution $z$-contrast scanning TEM images of lp-SNO, mp-SNO and hp-SNO are shown in Fig. 3a–c. The global arrangement of the atomic structure of lp-SNO (Fig. 3a) exhibits a long-range perovskite structure[13], whereas those of mp-SNO and lp-SNO

(Fig. 3b,c) exhibit a short-range perovskite structure interspersed with high densities of ordered {101} and {−101} extra oxygen planes that occur every few unit cells (shown in high magnification in Fig. 3d). Their position-averaged atomic structure (Fig. 3e) matches with the known extra oxygen plane structure of bulk $SrNbO_{3.4}$ (ref. 17) shown overlaid onto the image (see Supplementary Note 1 for details). The density of these planes qualitatively appears to increase with oxygen deposition pressure (Fig. 3a–c), and quantitatively (see Supplementary Figs 8 and 9 for details), there is a fourfold increase from mp-SNO to hp-SNO. This is consistent with an increased incorporation of extra oxygen with higher oxygen deposition pressure.

**Thickness-dependent study of correlated plasmons.** To examine size-dependent effects on the properties of the correlated plasmons, hp-SNO films with thinner thicknesses of $\sim 81$, $\sim 52$ and $\sim 20$ nm are also deposited and studied using SE, and the results are shown in Fig. 4 (see also Supplementary Figs 10 and 11 for more details). Interestingly, both the energy and number of excited correlated plasmons in hp-SNO change when the film thickness changes. In particular, the number of correlated plasmon peaks decreases as the film thickness decreases. When the film thickness decreases further to $\sim 20$ nm, the correlated plasmon disappears (Fig. 4d). This means that besides using the oxygen deposition pressure, the correlated plasmons can also be tuned by varying the thickness of the hp-SNO film. This is again fundamentally different from conventional plasmons where only the plasmon energy is changing when the metallic nanoparticle size is changed[3].

The disappearance of correlated plasmons in the $\sim 20$ nm hp-SNO film is particularly interesting. To investigate this further, we study the $\sigma_1(\omega)$ and $W$ of the thickness-dependent hp-SNO films shown in Fig. 5. It can be seen that as the film thickness decreases, the spectral weight of the first interband region ($>4.3$ eV), $W_3$, increases. This means that these thinner

hp-SNO films have less wide-range spectral weight transfer that signifies their correlation strength[11,24,30–35]. In particular, the $W_3$ of the $\sim 20$ nm hp-SNO film is almost as high as that of the metallic mp-SNO (Figs 2b and 5b), which means that the $\sim 20$ nm hp-SNO is most likely a weakly-correlated system. This further emphasizes the connection between the correlated plasmons, electronic correlation and the film dimensionality: as the film becomes thinner, the electronic correlation and thus the correlated plasmons also become weaker and ultimately vanish.

One possible reason for this thickness-dependent behaviour of the correlated plasmons might be due to the role of lattice mismatch. From Supplementary Fig. 1, it can be seen that the in-plane lattice constant of the $Sr_{1−x}NbO_{3+\delta}$ films is 4.04 Å, while the lattice constant of the $LaAlO_3$ substrate is 3.791 Å (ref. 36), which means that there is a relatively large lattice mismatch of $\sim 6.6\%$ between the films and the substrate. In thinner films, this large lattice mismatch should play a more significant role in influencing the electronic band structure of the whole film due to their relative thinness. The large lattice mismatch may reduce the unscreened Coulomb interactions between the electrons, resulting in less correlations and correlated plasmon excitations in thinner films. On the other hand, the role of lattice mismatch should be more minimized in thicker films, because their relative thickness should allow them to have more complete relaxations, leading to stronger correlations and more correlated plasmon excitations compared to thinner films.

**Reproducibility of correlated plasmons.** To test the reproducibility of the correlated plasmons, a second batch of pressure-dependent $Sr_{1−x}NbO_{3+\delta}$ films are deposited and measured using SE, and the analyses and results are shown in Supplementary Figs 12–17. The thickness of each film in this batch is kept within a narrow range of $\sim 159$–182 nm to minimize thickness-dependent effects (see Supplementary Table 1). From Supplementary Fig. 14, it can be seen that,

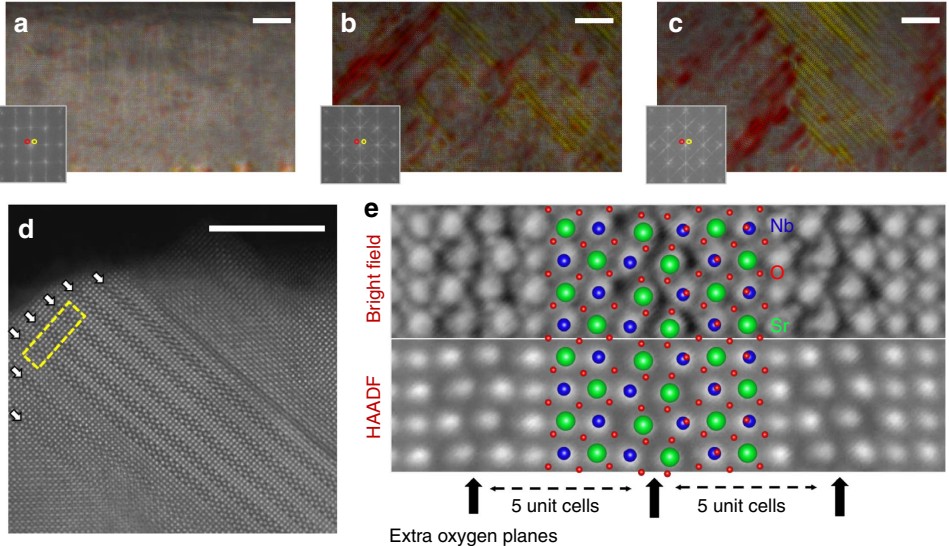

**Figure 3 | Atomic resolution scanning TEM of pressure-dependent $Sr_{1−x}NbO_{3+\delta}$ films.** (a–c) Atomic-resolution high-angle annular dark-field (HAADF) scanning TEM (STEM) images are shown for lp-SNO ($5 \times 10^{-6}$ Torr), mp-SNO ($3 \times 10^{-5}$ Torr) and hp-SNO ($1 \times 10^{-4}$ Torr) film, respectively. White scale bars, 8 nm. Arrays of {101} and {−101} extra oxygen planes manifest as superlattice peaks/streaks in the fast Fourier transform (FFT) patterns (inset). Bragg filtering is used to colorize the (−101) (red) and (101) (yellow) extra oxygen planes according to the respective highlighted regions of the FFT. (d) A higher magnification atomic-resolution HAADF image of hp-SNO depicts one such region of extra oxygen planes (arrows). (e) Averaged cross-section images corresponding to the template region highlighted in d are shown for both bright-field (inverted) and HAADF STEM images. The extra oxygen plane structure of $SrNbO_{3.4}$ (ref. 17) is overlaid (Sr = green, Nb = blue, O = red) and clearly matches the observed HAADF and bright-field atomic positions.

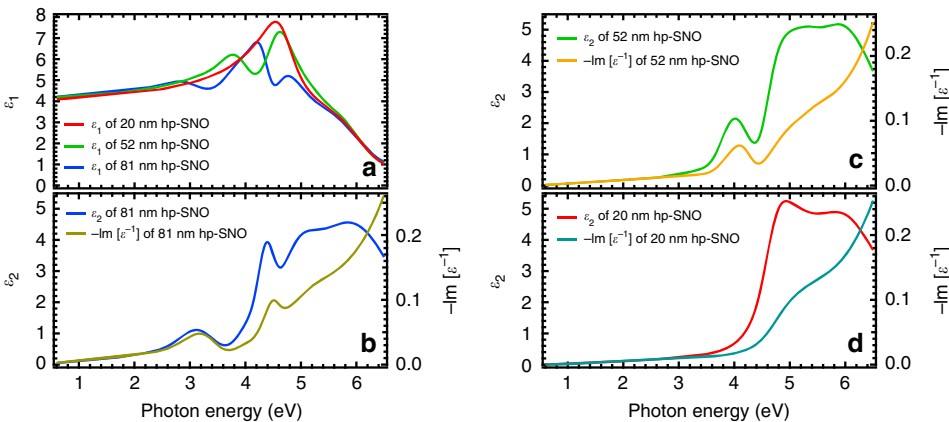

**Figure 4 | Complex dielectric function and loss function spectra of hp-SNO films with varying thicknesses.** (**a**) Real part of complex dielectric function, $\varepsilon_1(\omega)$, of hp-SNO ($1 \times 10^{-4}$ Torr) films with varying thicknesses. (**b**) Imaginary part of complex dielectric function, $\varepsilon_2(\omega)$, and loss function, $-\text{Im}[\varepsilon^{-1}(\omega)]$, of $\sim 81$ nm hp-SNO film. Here, two correlated plasmon peaks are observed at $\sim 3.2$ and $\sim 4.4$ eV. (**c**) The $\varepsilon_2(\omega)$ and $-\text{Im}[\varepsilon^{-1}(\omega)]$ of $\sim 52$ nm hp-SNO film. Here, the only correlated plasmon peak is observed at $\sim 4.0$ eV. (**d**) The $\varepsilon_2(\omega)$ and $-\text{Im}[\varepsilon^{-1}(\omega)]$ of $\sim 20$ nm hp-SNO film. No correlated plasmon peak is observed in this very thin film. For reference, the standard $\sim 168$ nm thick hp-SNO film has three correlated plasmon peaks.

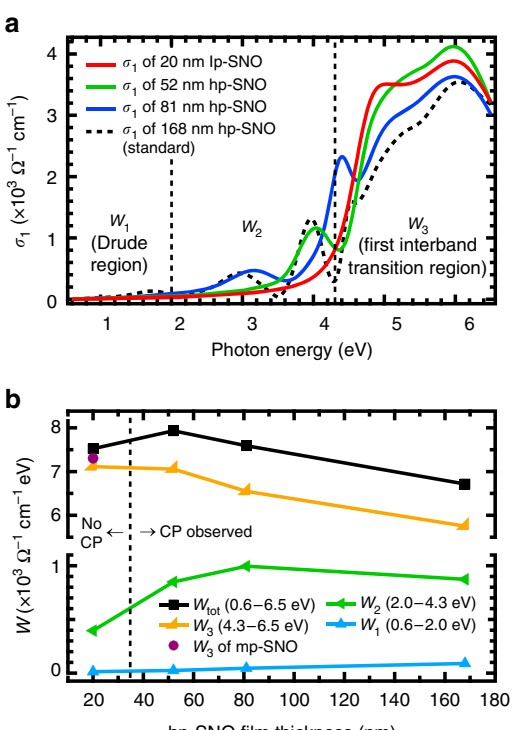

**Figure 5 | Optical conductivity spectra and spectral weight of hp-SNO films with varying thicknesses.** (**a**) The optical conductivity, $\sigma_1(\omega)$, of hp-SNO ($1 \times 10^{-4}$ Torr) films with varying thicknesses. The $\sigma_1(\omega)$ of the standard $\sim 168$ nm hp-SNO is also shown for reference. (**b**) Evolution of spectral weight, $W$, of three energy regions: $W_1$ (0.6–2.0 eV), $W_2$ (2.0–4.3 eV) and $W_3$ (4.3–6.5 eV) across hp-SNO films with varying thicknesses. The $W_{tot}$ is the total $W$ from 0.6 to 6.5 eV. The $W_3$ of mp-SNO (purple filled circle) is also shown for comparison. The abbreviation 'CP' stands for 'correlated plasmon'.

in general, the behaviours of the correlated and conventional plasmons as well as excitonic peaks in this batch are very similar to what was observed in the main-batch films shown in Fig. 1 (see Supplementary Note 4 for details). Furthermore, Supplementary Fig. 16 also shows that the pressure-dependent MIT is reproducible as well, and that the films deposited under higher oxygen pressures consistently have the signature of

strong electronic correlations. This indicates that the correlated plasmons are indeed reproducible, and its pressure-dependent behaviour can be reproduced consistently within similar thicknesses.

## Discussion

The presence of extra oxygen planes in hp-SNO and their absence in lp-SNO accompanied by remarkable changes in $\varepsilon(\omega)$ and $\sigma_1(\omega)$ spectra indicate that the interplay between extra oxygen planes and electronic correlations plays important roles in the plasmonics excitations and MIT between the films. For lp-SNO, its negative $\varepsilon_1(\omega)$ value below $\sim 1.9$ eV, strong Drude response and no excitonic peak indicate that the Coulomb interactions between its Nb-$4d$ electrons as well as between its electrons and holes are screened, making it a weakly-correlated metal similar to $Sr_{1-x}NbO_3$ (refs 14–16). Meanwhile, non-negative $\varepsilon_1(\omega)$ value, wide-range spectral weight transfer and presence of excitonic signature in thick ($\sim 168$ nm) hp-SNO indicate the Coulomb interactions among its Nb-$4d$ electrons as well as between its electrons and holes are unscreened, which leads to strong electronic correlation. In this regard, the extra oxygen planes in the thick hp-SNO act as high-potential walls that prevent the Nb-$4d$ electrons from hopping across the planes, confining them and making them feel stronger Coulomb repulsions. This changes their behaviour from itinerant to localized and transforms the system into a strongly-correlated Mott-like insulator[11].

Furthermore, since the extra oxygen planes are embedded throughout the volume of the hp-SNO film, the correlated plasmons disappear in the very thin hp-SNO film, and as hp-SNO itself is a Mott-like insulator with little to no free electrons at its surface, the observed correlated plasmons are most likely bulk plasmons instead of surface plasmons. The occurrence of the extra oxygen planes every few unit cells (every 5 unit cells for $SrNbO_{3.4}$) also means that the correlated plasmons have nanometre-spaced ($\sim 2$ nm for $SrNbO_{3.4}$) confinement in the film (see Supplementary Fig. 1 for the films lattice constants).

The electronic confinement is further supported by density functional theory (DFT) calculations shown in Fig. 6a,b. The calculations show that the presence of oxygen planes in $SrNbO_{3.4}$, which shares similar extra oxygen plane structure with hp-SNO, can indeed induce the electronic confinement on Nb-$4d$ electrons (Fig. 6b), consistent with previous report[16] (see also Supplementary Fig. 18 for details). This confinement is not observed in $SrNbO_3$ (Fig. 8a), which has no extra oxygen planes

similar to lp-SNO. In the following discussion, we model and explain the conventional and correlated plasmons of the thick films based on this electronic confinement induced by the extra oxygen planes.

Conventional plasmon can be classically described using the Drude model[3] as a collective oscillation of free charges against a positively charged ionic background (Fig. 6c). The theoretically calculated $\varepsilon(\omega)$ and LF spectra of lp-SNO using Drude model (Fig. 6e,f) agree very well with the experimental data below the bandgap. The theoretical LF of $SrNbO_3$ calculated using DFT with random phase approximation (RPA) method is also able to resemble qualitatively the experimental LF of lp-SNO (see Supplementary Fig. 19a and Supplementary Note 5 for details). This is not surprising because lp-SNO is a metal with little to no correlations and thus can be modelled using DFT-based calculations.

On the other hand, the theoretical LF of $SrNbO_{3.4}$ calculated using DFT and RPA is not able to resemble the experimental LF of thick hp-SNO, particularly the three correlated plasmons peak that we observe in thick hp-SNO (Supplementary Fig. 19b). This is because hp-SNO is a correlated system, which cannot be properly treated using DFT. Thus, to model the correlated plasmons of thick hp-SNO, we instead use a phenomenological model in which the effective Coulomb interactions between neighbouring correlated electrons is modelled as that of an elastic spring, at least in the low wave vector (or momentum transfer,

$|\mathbf{q}| \to 0$) limit. In this model (Fig. 6d and Supplementary Fig. 20), we qualitatively treat plasmonic oscillations of the correlated electrons as those of a chain of coupled harmonic oscillators bounded by two oxygen walls (where the walls are the extra oxygen planes that confine the Nb-4d electrons). Because the correlated electrons might have different effective masses and charges from bare electrons due to correlation effects[12], in the calculations we vary these along with the number of oscillators in one chain to find the set of parameters that can best describe the experimental data (see Supplementary Fig. 21 and Supplementary Note 3 for details).

From the calculation results (Fig. 6e,g), we find that the coupled oscillator model is able to resemble qualitatively the experimental $\varepsilon(\omega)$, LF and correlated plasmons of thick hp-SNO below the bandgap when the chain between the oxygen walls contains seven oscillators. To get the correct magnitude of $\varepsilon(\omega)$, the effective masses of the quasi-electrons are set to $\sim 25 m_e$, suggesting that the masses are heavily renormalized due to electronic correlation. Interestingly, we also find that to match the intensity trend of the correlated plasmon peaks, the quasi-electrons need to have alternating charges, that is, the quasi-electrons nearest to the walls have $-e$ charge, their immediate neighbours $+e$ charge, and so on, suggesting that the originally itinerant electrons have transformed into quasi-electrons and holes distributed alternatingly along the chain due to correlation. This is consistent with previous report

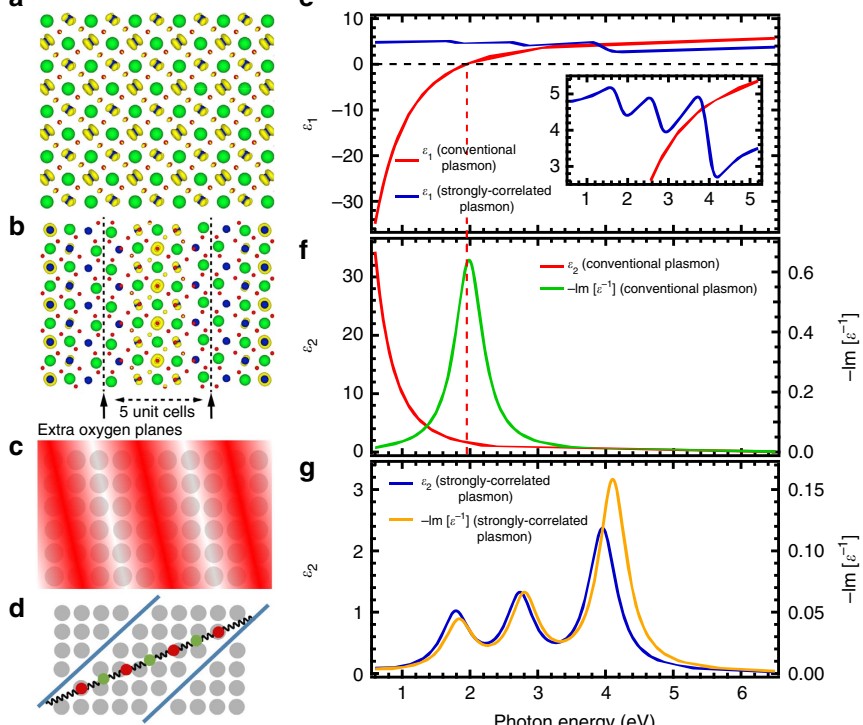

**Figure 6 | Coupled harmonic oscillator model of correlated plasmons.** (**a**) Theoretical Nb-4d electron density iso-surface of $SrNbO_3$ superimposed on $SrNbO_3$ crystal structure (Sr = green, Nb = blue, O = red, iso-surface = yellow). (**b**) Theoretical Nb-4d electron density iso-surface of $SrNbO_{3.4}$ superimposed on $SrNbO_{3.4}$ crystal structure. Black dashed lines denote the oxygen walls. The Nb-4d electrons occupy only the middle three Nb planes, while those close to the oxygen walls are depleted of electrons. (**c**) Illustration of conventional plasmon. Red wave pattern represents the oscillation of free charges, while grey spheres denote the underlying positively charged Nb ionic background. (**d**) Illustration of coupled harmonic oscillator model of correlated plasmon, showing the case where there are seven oscillators in one chain bounded by two oxygen walls (the coupled seven-oscillator model). Red and green spheres represent the renormalized quasi-electrons coupled by spring-like interaction, while blue lines represent the oxygen walls. (**e**) Calculated real part of complex dielectric function, $\varepsilon_1(\omega)$, of conventional plasmon using Drude model and that of correlated plasmon using coupled seven-oscillator model. Red dashed line indicates the zero-crossing of $\varepsilon_1(\omega)$ calculated using Drude model. Inset shows parts of spectra zoomed in for clarity. (**f**) Calculated imaginary part of complex dielectric function, $\varepsilon_2(\omega)$, and loss function, $-\text{Im}\,[\varepsilon^{-1}(\omega)]$, of conventional plasmon using Drude model. (**g**) Calculated $\varepsilon_2(\omega)$ and $-\text{Im}\,[\varepsilon^{-1}(\omega)]$ of correlated plasmon using coupled seven-oscillator model.

of possible formation of charge density wave in $SrNbO_{3.4}$ (refs 18,19).

The estimated heavy effective mass and the alternating charge arrangement of the quasi-electrons should also affect the dephasing time of the correlated plasmons. The heavy effective mass of the quasi-electrons can make the confined electrons (and thus the correlated plasmons) much more inertial against scattering, which can significantly enhance the correlated plasmon dephasing time. Furthermore, the redistributed charge arrangement would also alter their scattering profile to be different from that of nearly free bare electrons in metals.

In conclusion, we have demonstrated that depending on the electronic correlation strength, different types of plasmons can be reproducibly excited in $Sr_{1-x}NbO_{3+\delta}$ films. In weakly correlated metallic films, conventional plasmons dominate. Meanwhile, in strongly correlated Mott-like insulating films, low-loss correlated plasmons, which have fundamentally different properties and origins from conventional plasmons, are excitable instead. These correlated plasmons arise from the nanometre-spaced confinement of extra oxygen planes in the film and, in general, can be tuned by changing the electronic correlation strength. This can be accomplished by changing the extra oxygen plane density, for example, by varying the oxygen pressure during film deposition, and by changing the thickness of the films.

## Methods

**Materials preparations and characterizations.** The $Sr_{1-x}NbO_{3+\delta}$ films are deposited on (001) $LaAlO_3$ substrates by pulsed-laser deposition. The laser used is a Lambda Physik Excimer KrF ultraviolet laser with a wavelength of 248 nm, energy density of $2\,Jcm^{-2}$, and pulse frequency of 5 Hz. The three thick main-batch films are deposited at 750 °C and oxygen partial pressures of $5 \times 10^{-6}$, $3 \times 10^{-5}$ and $1 \times 10^{-4}$ Torr, respectively. The pulsed-laser deposition target is prepared by solid reactions of $Sr_4Nb_2O_9$ precursor, Nb (Alfa Aesar, 99.99%, $-325$ meshes) and $Nb_2O_5$ (Alfa Aesar, 99.9985%, metals basis) powder mixtures with proper molar ratio. The precursor is prepared by calcining $SrCO_3$ (Alfa Aesar, >99.99%, metals basis) and $Nb_2O_5$ powder mixtures in a molar ratio of 4:1. The calcination and sintering is done in air and 5% $H_2$–Ar gas environment for 20 h at a temperature of 1,200 and 1,400 °C, respectively. Different film thicknesses could be achieved by varying the deposition time. Typically, $\sim 130$-nm-thick film could be obtained with half an hour deposition.

The crystal structures of the main-batch films are studied using high-resolution XRD (Bruker D8 with Cu-Kα1 radiation and wavelength of 1.5406 Å) together with the reciprocal space maps. The XRD results (Supplementary Fig. 1) show that the in-plane lattice constants of all three films are 4.04 Å, close to that of bulk $SrNbO_3$ (refs 13,14). The out-of-plane lattice constant of lp-SNO and mp-SNO is 4.10 Å, which indicates that these two films have a slight uniaxial anisotropy along the out-of-plane ([001]) direction. Meanwhile, the out-of-plane lattice constant of hp-SNO is 4.02 Å, closer to the values of the in-plane lattice constants. The electronic transport properties of the films are measured using Physical Properties Measurement System (Quantum Design Inc.). The Sr/Nb cationic ratio of the films is measured using Rutherford backscattering spectrometry, and analyses of the obtained spectra using the SIMNRA simulation software yield an Sr/Nb ratio of 0.94 (see Supplementary Note 6 for more details about the stoichiometry of the films).

**Spectroscopic ellipsometry.** SE measurements are performed with specular reflection geometry at room temperature from 0.6 to 6.5 eV using Woollam V-VASE ellipsometer at 50°, 60° and 70° incident angles from the sample normal. The beam spot size is $\sim 1$–3 mm. The resulting ellipsometric parameters of each sample, $\tan\Psi \exp(i\Delta) \equiv r_p/r_s$, where $r_P$ and $r_s$ are the p- and s-polarized component of the amplitude reflection coefficients, respectively[37], are analysed to extract the complex dielectric function, $\varepsilon(\omega) = \varepsilon_1(\omega) + i\varepsilon_2(\omega)$, of the films using the Woollam WVASE32 and Woollam CompleteEase softwares. Since the samples are strontium niobates thin films on $LaAlO_3$ substrate, they are modelled as a two-layer system[37]. The $\varepsilon(\omega)$ of the thin films is fitted using a combination of Drude[37] and Herzinger–Johs PSemi-Tri[38] oscillator functions, while the $\varepsilon(\omega)$ of the underlying $LaAlO_3$ substrate is modelled using Lorentz oscillator functions[37] (all oscillator functions are Kramers–Kronig transformable[39]). To account for the surface ($\sim 1$ nm) and interface ($\sim 3$ nm) roughnesses, a Bruggeman-mode effective medium approximation is used[40]. The fitting is performed until a least $\chi^2$ fit is achieved. Owing to the indications of possible anisotropy from XRD data (Supplementary Fig. 1), the SE data of the films is analysed using two modes: (1) isotropic mode, where $\varepsilon(\omega)$ along all directions is assumed to be the same, and (2) uniaxial anisotropic mode, where the $\varepsilon(\omega)$ along the out-of-plane

(extraordinary) direction is assumed to be different from the $\varepsilon(\omega)$ along the in-plane (ordinary) directions. The fitting analysis of the $LaAlO_3$ substrate is performed using isotropic mode and the resulting $\varepsilon(\omega)$ spectra are shown in Supplementary Fig. 2.

The SE data of lp-SNO and mp-SNO, as well as lp-SNO-2, mlp-SNO-2, mp-SNO-2 and mhp-SNO2, cannot be fitted properly using isotropic mode and can only be fitted well at all incident angles using uniaxial anisotropic mode (Supplementary Figs 3 and 12). This indicates that these conducting films have a slight uniaxial anisotropy along the out-of-plane direction, consistent with XRD data (Supplementary Fig. 1). The resulting ordinary (along in-plane directions) $\varepsilon(\omega)$ and LF, $-\text{Im}\,[\varepsilon^{-1}(\omega)]$, of lp-SNO and mp-SNO films are shown in Fig. 1, while their extraordinary (along out-of-plane direction) $\varepsilon(\omega)$ and LF spectra are shown in Supplementary Fig. 4. Meanwhile, the resulting ordinary $\varepsilon(\omega)$ and LF of the second-batch conducting films are shown in Supplementary Fig. 14, while their extraordinary components are shown in Supplementary Fig. 15. Since the features of extraordinary $\varepsilon(\omega)$ of lp-SNO and mp-SNO are similar to their ordinary $\varepsilon(\omega)$, only the ordinary $\varepsilon(\omega)$ of lp-SNO and mp-SNO are shown in Fig. 1 for clarity. The SE analyses also show that the thicknesses of lp-SNO and mp-SNO films are $\sim 196$ nm (with thickness non-uniformity of $\sim 11\%$) and $\sim 218$ nm (with thickness non-uniformity of $<5\%$), respectively. The thicknesses of the second-batch films are shown in Supplementary Table 1.

Meanwhile, the SE data of hp-SNO, as well as hp-SNO-2 and hrp-SNO-2, can be fitted well at all incident angles using the isotropic mode (Supplementary Figs 5 and 13) and analysis using anisotropic mode yields the same ordinary and extraordinary $\varepsilon(\omega)$ and LF spectra of the film, indicating that hp-SNO is optically isotropic and it has little to no anisotropy (see Supplementary Note 7 for more details). This is consistent with XRD data (Supplementary Fig. 1), which shows that the in-plane and out-of-plane lattice constants of hp-SNO are much closer to its in-plane lattice constants as compared to those of lp-SNO and mp-SNO. The resulting $\varepsilon(\omega)$ and LF spectra of hp-SNO are shown in Fig. 1, while those of hp-SNO-2 and hrp-SNO-2 are shown in Supplementary Fig. 14. The SE analysis also shows that the thickness of the thicker hp-SNO film is $\sim 168$ nm (with thickness non-uniformity of $\sim 34\%$). The SE data of thinner hp-SNO films ($\sim 81$, $\sim 52$ and $\sim 20$ nm) is similarly analysed using isotropic mode (Supplementary Fig. 10).

From the obtained $\varepsilon(\omega)$, the refractive index ($n(\omega)$), extinction coefficient ($\kappa(\omega)$), normal-incident reflectivity ($R(\omega)$), and absorption coefficient ($\alpha(\omega)$) of the films can be extracted using the Fresnel equations and the following relations:

$$n(\omega) = \sqrt{\frac{1}{2}\left[\sqrt{\varepsilon_1^2(\omega) + \varepsilon_2^2(\omega)} + \varepsilon_1(\omega)\right]}, \quad (1)$$

$$\kappa(\omega) = \sqrt{\frac{1}{2}\left[\sqrt{\varepsilon_1^2(\omega) + \varepsilon_2^2(\omega)} - \varepsilon_1(\omega)\right]}, \quad (2)$$

$$R(\omega) = \frac{[n(\omega) - 1]^2 + \kappa^2(\omega)}{[n(\omega) + 1]^2 + \kappa^2(\omega)} \quad (3)$$

and

$$\alpha(\omega) = \frac{4\pi\kappa(\omega)}{\lambda}, \quad (4)$$

where $\lambda$ is the photon wavelength. The $n(\omega)$, $\kappa(\omega)$, $R(\omega)$ and $\alpha(\omega)$ of lp-SNO, mp-SNO and $\sim 168$ nm hp-SNO are shown in Supplementary Fig. 6, while the $n(\omega)$, $\kappa(\omega)$, $R(\omega)$ and $\alpha(\omega)$ of thin hp-SNO films are shown in Supplementary Fig. 11. It should be noted that since the extraordinary $\varepsilon(\omega)$ of lp-SNO and mp-SNO are very similar to their ordinary components (Supplementary Fig. 4), for simplicity only the ordinary $\varepsilon(\omega)$ of lp-SNO and mp-SNO are used to calculate their respective $n(\omega)$, $\kappa(\omega)$, $R(\omega)$ and $\alpha(\omega)$.

**Conventional plasmon dephasing time.** Based on $\varepsilon(\omega)$ and LF, the free electron scattering, $1/\tau$, can be estimated by at least two methods. In the first method, by using the Drude model (Supplementary equation 6b), $1/\tau$ can be extracted from the blueshift between the plasmon energy, $\omega_P$, and the zero-crossing of $\varepsilon_1(\omega)$, $\omega_{\varepsilon_1=0}$,

$$\frac{1}{\tau} = \sqrt{\omega_P^2 - \omega_{\varepsilon_1=0}^2}. \quad (5)$$

In the second method, the free electron scattering can be estimated from the FWHM, $2\hbar\Gamma$, of the conventional plasmon peak taken from the LF spectra. The free electron scattering contributes to the dephasing of the conventional plasmon, meaning that the conventional plasmon dephasing time[3], $T_2$, can be obtained using

$$T_2 = \frac{1}{\Gamma} = 2\tau. \quad (6)$$

**Optical conductivity and spectral weight.** Since the $\varepsilon(\omega)$ of lp-SNO and mp-SNO, as well as lp-SNO-2, mlp-SNO-2, mp-SNO-2 and mhp-SNO-2, are slightly anisotropic, their optical conductivity, $\sigma_1(\omega)$, and spectral weight, $W$, are

also anisotropic because they are both derived from $\varepsilon(\omega)$. The ordinary (o) $\sigma_1(\omega)$ of the films are shown in Fig. 2a (main batch) and Supplementary Fig. 16a (second batch), while the corresponding extraordinary (e) components are shown in Supplementary Fig. 7a (main batch) and 17a (second batch). Meanwhile, the evolutions of the ordinary ($W_o$) and extraordinary ($W_e$) components of spectral weight across the three main-batch films are shown in Supplementary Fig. 7b,c, respectively, while for the second-batch films they are shown in Supplementary Fig. 17b,c, respectively. The axis-averaged spectral weight shown in Fig. 2b and Supplementary Fig. 16b is obtained using $W = \frac{2}{3}W_o + \frac{1}{3}W_e$, since the anisotropy of the conducting films is uniaxial with two ordinary in-plane axes and one extraordinary out-of-plane axis. The $W$ is proportional to the number of charges participated in the optical excitation, thus the $f$-sum rule is a charge conservation rule. If the $W$ of a particular energy region increases, it has to be compensated by an equivalent decrease of the $W$ of another energy region, and *vice versa*.

**Transmission electron microscopy.** The main-batch $Sr_{1-x}NbO_{3+\delta}$ thin films are prepared for TEM measurements via the lift-out method using an FEI Strata 235 dual beam focused ion beam. The samples are then locally ion milled using a Fischione 1040 Nanomill at 900 eV and then 500 eV to remove the focused ion beam damaged surfaces. The TEM and selected-area electron diffraction (SAED) are performed on a JEOL 3010 operating at 300 kV. Zone axis SAED patterns are collected for all three films (Supplementary Fig. 8) back-to-back using identical SAED apertures, magnification and beam illumination (beam fully spread). Distinct Bragg peaks are visible for the $Sr_{1-x}NbO_{3+\delta}$ films and LaAlO₃ substrate in both the out-of-plane (00*l*) and in-plane (*h*00) directions indicating at least partial strain relaxation for all three cases. Superlattice reflections along the (101) and (−101) axis surround the main SrNbO₃ Bragg peaks in the mp-SNO and hp-SNO SAED patterns, forming X-shaped streaks, due to diffraction from the extra oxygen planar defects. A magnified region around the (100) Bragg peaks for each sample is shown at the bottom of Supplementary Fig. 8 plotted on identical, colorized, intensity scales. To isolate the superlattice reflections the main SrNbO₃ and LaAlO₃ peaks are masked out. For consistency the mask locations are determined by a least-squares fit of a uniform grid to the SrNbO₃ and LaAlO₃ Bragg peaks whose positions were determined by Gaussian fit. The background is subtracted as the radial median value from the centre of the diffraction pattern. To quantify the increase in the superlattice peak intensity, the mean value was calculated within the X-shaped region shown overlaid on the (100) peak images in Supplementary Fig. 8 while the mean value outside is taken as the zero value. The resulting value is then normalized by the sample thickness, $t/\lambda$ (that is, thickness measured in units of the inelastic mean free electron scattering distance, $\lambda$, see below and Supplementary Fig. 9) giving a superlattice intensity measure of 19.16, 92.91 and 317.04 for lp-SNO, mp-SNO and hp-SNO, respectively. From this increase in the superlattice reflection intensity from lp-SNO, to mp-SNO, to hp-SNO, it is straightforward to infer an increase in oxygen defect plane density and an increase in oxygen incorporation.

Atomic resolution scanning TEM and energy-filtered TEM are performed on the TEAM0.5, a Cs aberration-corrected FEI Titan operating at 300 kV. For high-resolution scanning TEM, simultaneous images are collected using a high-angle annular dark-field (HAADF) detector and a bright-field detector. The former, used in all panels of Fig. 3, produces $Z$-contrast images with intensities approximately proportional to the square of the atomic number ($Z$) and atomic nuclei appearing bright. The bright-field images, used in the top half of Fig. 3e, are useful for detecting light elements like oxygen as demonstrated by the visibility of oxygen sites in the planar defects. The bright-field image contrast in Fig. 3e has been inverted to give it a bright-atom appearance like the HAADF. Thickness mapping of the TEM cross-sections along the electron beam axis (Supplementary Fig. 9) is performed using energy-filtered TEM. To calculate $t/\lambda$ thickness maps, pairs of images are collected consisting of an unfiltered and an energy-filtered image using a 7 eV slit centred on the zero-loss peak.

**Density functional theory calculations.** The atomic and electronic structure of $SrNbO_{3+\delta}$ compounds are computed by spin-polarized DFT calculations using the Perdew–Burke–Ernzerhof (PBE96)[41] exchange-correlation potential and the projector-augment wave (PAW) method[42] as implemented in the Vienna *ab initio* simulation program[43]. In these calculations, Sr-4*s*4*p*5*s*, Nb-4*p*5*s*4*d* and O-2*s*2*p* orbitals are treated as valence states, employing the PAW potentials labelled as Sr_sv, Nb_pv and O in the Vienna *ab initio* simulation program PBE library. The cutoff energy for the plane-wave basis set is set to 450 eV, and DFT + U approach[44] is employed to treat the Nb-4*d* orbitals occupied in the Nb⁴⁺ ions, with the value of $U - J$ set to 4 eV. SrNbO₃, SrNbO₃.₃₃, SrNbO₃.₄ and SrNbO₃.₅ are modelled by supercells containing, respectively, 20 atoms with space group *Pnam*, 64 atoms with space group *Ccmm*, 54 atoms with space group *Pnnm* and 44 atoms with space group *Cmc2*. In the structural relaxations, $8 \times 8 \times 4$, $1 \times 4 \times 6$, $1 \times 4 \times 6$ and $1 \times 4 \times 6$ $k$-point meshes are employed for SrNbO₃, SrNbO₃.₃₃, SrNbO₃.₄ and SrNbO₃.₅, respectively. The Nb-4*d* valence electron densities of SrNbO₃ and SrNbO₃.₄ are shown in Fig. 6a,b, while those of SrNbO₃.₃₃ and SrNbO₃.₅ are shown in Supplementary Fig. 18 and discussed in Supplementary Note 8. The valence electron densities are calculated using electron states with energies ranging from the bottom of the Nb conduction band up to the Fermi level. The samples used in experiments are characterized to have 6% Sr vacancies, and to account for this we estimate that the main effect of Sr vacancies is to shift the Fermi level downward in

energy without changing the shape of the valence and conduction bands significantly. For SrNbO₃.₄ (Fig. 6b), the Nb-4*d* electrons occupy only the middle three Nb planes, while those close to the oxygen walls are depleted of electrons. For SrNbO₃ (Fig. 6a), no such confinement is observed.

The theoretical LF spectra of SrNbO₃ and SrNbO₃.₄ are calculated as the inverse of the macroscopic complex dielectric function in RPA[45]. The LF calculations are performed on top of previous DFT + PAW ground state calculations with an increase of the $k$-point mesh grid of up to $16 \times 16 \times 1$ and by including 350 bands. In RPA calculations, any short electron–hole interactions are neglected and thus the calculations can only result in conventional plasmons but not correlated plasmons. The conventional plasmons themselves are determined by the zeroes of the real part of the complex dielectric function (Supplementary equation 9). The resulting LF spectra of SrNbO₃ and SrNbO₃.₄ calculated using this method are shown in Supplementary Fig. 19.

It should be noted that DFT has a limitation in calculating correlated electron systems because it cannot take into account correlation effects properly, in particular the spectral density transfers relevant for Mott physics. This is particularly seen in its inability to calculate LF spectrum that can resemble the experimental LF of the correlated hp-SNO film (Supplementary Fig. 19b). Thus, in the future more rigorous theoretical approaches are needed to shed a more complete picture of the electron confinement and its associated correlated plasmon excitations.

**Conventional and correlated plasmons theoretical calculations.** The theoretical $\varepsilon(\omega)$, LF, and conventional plasmon of lp-SNO shown in Fig. 6e,f are calculated using the established Drude model[3] with the following parameters: $m^\star = m_e$, $n = \frac{1}{V_{cell}}$, $\omega_p = 2$ eV, $\varepsilon(\infty) = \frac{ne^2}{m\omega_p^2}$ and $1/\tau = 0.5$ eV, where $m^\star$, $V_{cell}$, $\omega_p$ and $\tau$ is the electron effective mass, the volume of film unit cell, the plasmon frequency and the lifetime in which electrons can move without experiencing a scattering process, respectively. The $V_{cell}$ is calculated from the lattice constants obtained from XRD (Supplementary Fig. 1) using cubic perovskite structure. With this choice of parameters, the calculation results are in a very good agreement with experimental data up to 4 eV, below the onset energy of the interband transitions, confirming that lp-SNO is a metal with conventional plasmonic characteristics.

Meanwhile, the theoretical $\varepsilon(\omega)$, LF and correlated plasmon of hp-SNO shown in Fig. 6e,g are calculated using the coupled harmonic oscillator model as a qualitative approach in the low wave vector limit. The thought process regarding this model can be elaborated as follows. In non- or weakly correlated metals, the electron–electron repulsions are mostly screened. Thus, when conventional plasmons are excited, the electrons oscillate as a whole against the positive ionic background, and not against each other since the electron–electron interactions are largely suppressed. In this case, the restoring force is only due to the Coulomb attraction with the positive ionic background (Fig. 6c and Supplementary equation 1), resulting in one bulk plasmon frequency that depends only on the free electron density and effective mass. On the other hand, in strongly correlated materials, electron–electron repulsions are unscreened. Thus, when correlated plasmons are excited, the correlated electrons should also oscillate against each other since each electron can now feel the Coulomb repulsions from neighbouring electrons. In this case, the restoring force would also include the electron–electron repulsions, which turns the plasmonic oscillation into a many-body motion with multiple possible natural frequencies (Supplementary equation 10).

Based on this, the correlated plasmon oscillations of Nb-4*d* electrons in hp-SNO are modelled as that of a chain of coupled harmonic oscillators bounded by two oxygen walls (Fig. 6d and Supplementary Fig. 20). As a first approximation, the electron–electron interactions are modelled as that of an elastic spring, with the spring constant representing the strength of the electronic correlation. Furthermore, due to correlation effects, the effective masses and charges of the Nb-4*d* electrons might become renormalized[12] and different from those of bare electron. Thus, in the calculations the effective masses and charges of the correlated electrons along with the number of oscillators in one chain ($N$) are fine-tuned as follows to find the set of parameters that can best match the experimental data, especially the three correlated plasmon peaks at ~1.7, ~3.0 and ~4.0 eV (Fig. 1a,d).

First, to tune the peak positions, we set an energy scale of about the lowest energy of the three plasmon peaks, that is, $\omega_0 = 2.0$ eV. We also set a scale for the effective mass values to be $m^\star = Zm_e$, and a scale for the spring constant values to be $k_0 = m^\star\omega_0^2$. Here, $Z$ is the mass renormalization factor that we initially set to be equal to 1 and adjust it as necessary. As initial guesses, we set $m_1 = m_2 = \cdots = m_N = m^\star$ and $k_1 = k_2 = \cdots = k_{N+1} = k_0$, where the indices, $i = 1$, 2, ..., are according to the oscillator arrangement in Supplementary Fig. 20. Fine-tuning of the peak positions is done by adjusting individually each $k_i(m_i)$ slightly away from $k_0(m^\star)$. The $Z$ value is also fine-tuned so that the magnitudes of the calculated $\varepsilon_1(\omega)$ and $\varepsilon_2(\omega)$ match closely with experimental data, and the optimum $Z$ value is found to be ~25.

Second, to tune the peak intensity trend which increases as energy increases (see Fig. 1d), we fine-tune the effective charges of the correlated electrons. If the effective charges are set to be all the same as that of bare electron, the calculation results in a peak intensity trend that decreases as energy increases, which is the opposite trend compared to what was observed experimentally. This happens because the slower vibration mode corresponds to higher polarization, which gives

higher peak intensity as energy increases. To improve this, we find that the increasing intensity trend can only be achieved if the charges are assumed to be distributed with alternating signs, that is, $q_1 = -e$, $q_2 = +e$, $q_3 = -e$ and so on.

Third, the number of plasmonic peaks appearing in the theoretical $\varepsilon(\omega)$ and LF of hp-SNO is tuned by varying the number of oscillators in one chain. In analogy with classical mechanics, an $N$-oscillator system has $N$ natural oscillation modes. Since the number of observed plasmonic peaks in Fig. 1d is three, this gives an indication that $N$ should be an odd number. We start with the case of $N = 3$. A three-body oscillator have three intrinsic oscillation modes, but there is one mode that has zero polarization. This zero-polarization mode cannot show up in $\varepsilon(\omega)$, leaving only two non-zero peaks and inconsistent with the three peaks observed in Fig. 1d. Next, for $N = 5$ the five-body oscillator has five eigenmodes, but two of them have zero polarization, leaving three non-zero peaks surviving, which is what the experimental results require. The calculation results (Supplementary Figure 21) show that for $N = 5$ the peak positions and the increasing intensity trend are in a rough agreement with experimental results, but the calculated relative peak intensities are not very satisfactory because the first and the second peaks are too low compared to the third peak.

The calculation for the case of $N = 7$ is next performed with the same approach as discussed above. The results show that not only does it give three appearing peaks (Fig. 6e,g and Supplementary Fig. 21), with only an additional negligibly small peak close to 0 eV, but also it provides a peak intensity profile that much better resembles the experimental results. For this particular case, the parameter values are set as follows: $\omega_0 = 2.0$ eV, $Z = 25$, $\varepsilon_1(\infty) = 4.0$, $1/\tau = 0.5$ eV, $m_1 = m_2 = \cdots = m_N = Zm_e$, $k_1 = 0.4k_0$, $k_2 = 0.5k_0$, $k_3 = 1.0k_0$ and $k_4 = 1.1k_0$, and the rest of the spring constants follow a mirror symmetry, that is, $k_8 = k_1$, $k_7 = k_2$, $k_6 = k_3$ and $k_5 = k_4$.

The calculations are also performed for $N = 9$ and $N = 11$, although Supplementary Fig. 21 shows that these two cases result in too many peaks. In Fig. 1d it can be seen that, in addition to the three main peaks of $C_1$, $C_2$ and $C_3$, there is a small peak at $\sim 2.4$ eV that appears in both $\varepsilon(\omega)$ and LF spectra of thick hp-SNO, which means this peak is also a correlated plasmon peak. From the analysis of Supplementary Fig. 21, this small peak could come from the excitation of these higher-order modes (that is, $N = 9$ and/or $N = 11$), although its low intensity compared the three main peaks of $C_1$, $C_2$ and $C_3$ means that the seven-oscillator mode should still be the dominant mode of oscillation.

For more details, the in-depth theoretical calculations of the conventional plasmon of lp-SNO and correlated plasmons of hp-SNO are discussed further in Supplementary Note 3.

**Data availability.** The data that support the findings of this study are available from the corresponding authors on reasonable request.

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

## Acknowledgements

This work is supported by Singapore National Research Foundation under its Competitive Research Funding (NRF-CRP 8-2011-06), MOE-AcRF Tier-2 (MOE2015-T2-1-099), 2015 PHC Merlion Project and FRC. We thank Centre for

Advanced 2D Materials and Graphene Research Centre at the National University of Singapore to provide the computing resource. D.W., Y.Z., B.Y. and T.V. prepared high-quality samples and performed transport and structural measurements.

## Author contributions

T.C.A., D.S. and A.R. performed spectroscopic ellipsometry measurements and analysed the resulting spectra. D.W., Y.Z., B.Y. and T.V. prepared high-quality samples and performed transport and structural measurements. C.T.N., M.C.S. and A.M. performed high-resolution TEM measurements. T.C.A., M.A.M. and A.R. constructed the main theoretical model. M.A.M., Y.C., M.Y., T.Z., P.E.T., M.S., M.A. and A.R. performed theoretical calculations. M.R.M., M.B.H.B. and T.V. performed Rutherford backscattering measurements. A.R. and T.C.A. wrote the paper with input from all coauthors. A.R. and T.V. initiated and lead the project.

## Additional information

**Competing interests:** The authors declare no competing financial interests.

