## [Peer Review File · Nature Communications]

Reviewers' comments:

Reviewer #1 (Remarks to the Author):

A. Summary of the key results

The authors use principally ellipsometry to ascertain that they observe a new phenomenon that they termed "correlated plasmon." Although this could be a very interesting concept, there is not enough experimental evidence (that is experiments directly connected to understanding the said novel plasmonic properties) to justify these claims.

Moreover, the authors do not discuss the nature of the plasmonic resonance, such as the dephasing that are associated with such correlated plasmons.

B. Originality and interest: if not novel, please give references

Potentially novel, but not enough evidence.

C. Data & methodology: validity of approach, quality of data, quality of presentation

Experiments connected to exciting plasmons

(I) grating, Otto/Kretschmann configuration)

(II) Plasmonic size dependence

(III) Transmission/Reflection/Absorption measurement of the films

(IV) EELS experiment

are all lacking.

D. Appropriate use of statistics and treatment of uncertainties

Not enough samples, and systematic studies varying the film thicknesses or interlayers.

G. Not enough credit/references that combine the work related to plasmonic and Mott-effect.

Reviewer #2 (Remarks to the Author):

In the reviewed manuscript, the authors present an extensive and conclusive study of a new type of tunable correlated plasmons observed in a wide bandgap, Mott-like insulating oxide from the $\text{Sr}_{1-x}\text{Nb}_1-y\text{O}_{3+\delta}$ family. Due to the long-range Coulomb interactions, the electrons in the oxide are strongly correlated and the corresponding collective excitations of the correlated electrons result in multiple plasmon frequencies with low losses. The authors experimentally observed the correlated plasmons and bring us to a new plasmonic material in the visible-ultraviolet range, based on wide bandgap and strongly correlated materials, tackling one of the most challenging issues in plasmonics research. The authors interpret the physical origin of the observed tunable correlated plasmons in $\text{Sr}_{1-x}\text{Nb}_1-y\text{O}_{3+\delta}$ as the nanometer-spaced electronic confinement of extra oxygen planes resulting from crystal growth under high oxygen pressure, which is fully supported by the density functional theory calculations together with a coupled oscillator model.

I have only a few minor questions/comments for the authors to address before accepting this work for publication in Nature Communications.

(1) SNO films with different concentration of oxygen are produced and their crystallinity is characterized by TEM and powder X-rays. Is there any direct indication of the stoichiometric ratio for oxygen in SNO films? During growth, the crystal defects could be produced by the exchanged cations, such as vacancies. Can the authors estimate the effect of the crystal defects on the correlated plasmons?

(2) The authors should add the description of the growth mechanism for oxygen planes formation in hp-SNO films. Why can the oxygen atoms form pure planes without forming other atomic arrangement

manners?

(3) Although the oxygen planes can be clearly observed from the TEM images of hp-SNO, their global arrangement and detailed structures in the hp-SNO films remain unclear. Since the exact arrangement of oxygen planes is critical for the density functional theory calculations, the authors have to describe this aspect in great detail and specify the dimensionality of the space confinement on the carriers.

(4) The mode of correlated plasmons is dependent on the spaced confinement that can be tuned by changing the density of oxygen planes. I suggest the authors carrying out additional DFT calculations for a series of oxygen plane densities in hp-SNO films. This may help understand the exact role of oxygen planes in generating the observed tunable and correlated plasmons.

(5) From Fig. 1a-c, the lp-SNO and mp-SNO films show the large peaks of A'1 and B'1, respectively. These peaks coincide with the zero-crossing of $\epsilon_{-1}(\omega)$, indicating that the peak comes from a conventional plasmon resonance. However, both peaks A'1 and B'1, have a small blue-shift compared to the zero-crossing. This blue-shift is not discussed in the manuscript and is important for the conventional plasmon resonance.

Reviewer #3 (Remarks to the Author):

The paper is very interesting; indeed, to my knowledge there were no successful attempts to study experimentally plasmons in strongly correlated systems. As a first attempt of this kind, the work deserves to be published. Theoretical part is a bit naive; at the same time, it is known that in the limit of small wave vectors the classical model of coupled oscillators may be qualitatively correct. Of course, it cannot be sufficient to describe EELS with finite wave vector, and this should be said explicitly. At the same time, it would be nice to have a more detailed comparison with theoretical results (Ref.7); the most important in the theory is not that it can be a renormalization of parameters like effective mass etc. but that in principle several types of plasmon excitations are possible. It is not completely clear how the results of density-functional calculations (which do not take into account properly correlation effects, in particular, spectral density transfer relevant for Mott physics) are related to the experimental results. Currently, much better calculations can be done within, e.g., LDA+DMFT (dynamical mean field theory). I do not mean that this kind of calculations is obligatory for this particular paper, this is first of all an experimental work with some auxiliary theoretical material, but, at least, all restrictions of the method used in the calculations should be clearly formulated. In general, I believe this is an interesting and important work of a broad interest which may be suitable for publication in Nature Communications but I would recommend the authors to improve discussion to make theoretical part and its relevance for experiment more clear.

Reviewers' comments:

Reviewer #1 (Remarks to the Author):

Although I think the idea of correlated plasmonic is a potentially interesting one, the lack of clear direct evidence of correlation and more importantly how this is a plasmonic effect is now more striking with the size-dependent thin-film measurement. This is clearly not suitable for Nature Communications.

- Also, other groups have discussed correlations in regard to such Mott-type or exotic materials, such as the group by Basov or Haglund (VO₂), Zheludev (YBCO superconducting plasmonics) and Cavalleri (Cuprate superconductors for transient plasmonic response) to name a few.

- Importantly, with such complex structures such as the perovskite ones, it is known that their electronic and optical properties are also complex and arise from various band splitting, both in the conduction and valence bands. This often occurs right in the region shown by the authors, with no plasmonic effect or correlation effect.

This arises due to spin-orbit coupling, electron-phonon coupling, excitonic effects and anisotropic dielectric functions. For example, see the early work by Mitzi on layered perovskite or the more recent work by the photovoltaics community who are currently assessing the electronic structure of these various perovskite materials (e.g. Even or Fujiwara).

- The DFT and harmonic coupled oscillator are quite naïve regarding the complexity of the material system, as also mentioned by the other reviewers. As pointed out by the authors, DFT has difficulties capturing the correlation effects of various systems and whether bandgap calculation renders similar experimental results should at least be mentioned in the manuscript. Also, how do changes in the space group affect this analysis with such a small number of atoms in the supercell? As for the coupled oscillators, I still do not see how correlation connects to these experimental observations.

Other queries about the paper:

1. The non-systematic work with only three main samples that also have varying thicknesses (> 50 nm difference). The only thickness-dependent measurement carried out does not support their arguments of correlations. Instead, it supports the fact that there are various perovskite structures with varying optical/excitonic transitions. Also, other peaks - at 2.25 (between C1 and C2) and 6 eV (after C5) - are not addressed in the paper. There is also a peak between B3 and B4, albeit small potentially, that is not mentioned in the paper.

2. The author discusses the advantages of surface plasmon in the introduction as motivation, but later on proceed to discuss that this is only a bulk effect, with these peaks vanishing quite rapidly as the sample thickness is decreased.

3. What is the lattice mismatch between the various structures and the LCMO substrate?

4. From Fig3, it appears there are mixed states present (sample inhomogeneity) which result also in this anisotropic dielectric functions. How were these two samples (3b and 3c) modeled differently?

Reviewer #2 (Remarks to the Author):

Since the authors have adequately addressed my comments made in the first run of review, I am

ready to accept the revised version for publication in Nature Communications after the authors have made the following minor revision.

- Line 21, Page 2: note that localized surface plasmons have also been observed in the metallic state of VO₂ - a phase changing material and used to probe the role of defects in the phase transition of VO₂ nanoparticles, see Nano Letters 2012, 12, 780-786. The authors should cite this representative work in order to make the literature review more complete.

Reviewer #3 (Remarks to the Author):

I am satisfied by the changes in the manuscript made by the authors and have found their reply to the referee's criticism adequate and convincing. I think the paper may be published in the present form.

REVIEWERS' COMMENTS:

Reviewer #1 (Remarks to the Author):

Please accept the manuscript since they have addressed all lingering concerns.

Note: All reviewers comments/questions are in black. Our response to them and change made in manuscript are highlighted in blue.

Reviewers' comments:

Reviewer #1 (Remarks to the Author):

Comment R1.1: A. Summary of the key results

The authors use principally ellipsometry to ascertain that they observe a new phenomenon that they termed "correlated plasmon." Although this could be a very interesting concept, there is not enough experimental evidence (that is experiments directly connected to understanding the said novel plasmonic properties) to justify these claims.

Response R1.1: First of all, we would like to thank Reviewer #1 for his/her feedbacks on our manuscript. Below, we provide further evident and explanation on the observation of correlated plasmons in this system.

As discussed in the main article, we would like to highlight that this correlated plasmon phenomenon that we observe is new and fundamentally different from conventional plasmons as in metals. As we have discussed extensively in paragraph 7 (see previous main article), these correlated plasmons occur in a strongly-correlated Mott-like insulator, have multiple plasmon frequencies, occur in both $\varepsilon_2(\omega)$ and loss function ($-\text{Im}[\varepsilon^{-1}(\omega)]$) spectra and do not coincide with the zero crossing of $\varepsilon_1(\omega)$. These properties are very different from those of conventional plasmons that occur in weakly-correlated metals, only have one bulk

plasmon frequency, occur only in the loss function spectra but not in $\varepsilon_2(\omega)$ and coincide with the zero crossing of $\varepsilon_1(\omega)$.

Furthermore, the theoretical framework to explain these correlated plasmons is also still at the early stage of development, unlike conventional plasmons whose theory has already been very well established. For comparison, to our knowledge the first theoretical calculations, which explicitly include strong electronic correlations, to study plasmons in correlated systems (Ref. 7) was published only very recently on December 2014, while the Drude model used to explain metallic free electrons and conventional plasmons is already more than a century old. Thus, our result opens new research directions in both experiment and theory as we discuss in details below.

Comment R1.2: Moreover, the authors do not discuss the nature of the plasmonic resonance, such as the dephasing that are associated with such correlated plasmons.

Response R1.2: We would like to thank Reviewer #1 for this feedback about the plasmon dephasing that we did not explicitly discuss in the previous version of our manuscript.

Let's start our discussion by considering a conventional plasmon. Based on complex dielectric functions and loss function, there are at least two methods to estimate the *free-*electron scattering, $1/\tau$, where τ is the lifetime in which electrons can move without experiencing a scattering process, and thus the conventional plasmon dephasing time. In the first method, by using the Drude model (Supplementary Eq. 6b), the free electron scattering can be extracted from the blue shift between the plasmon energy, ω_p , and the zero crossing of $\varepsilon_1(\omega)$, $\omega_{\varepsilon_1=0}$,

$$\frac{1}{\tau} = \sqrt{\omega_p^2 - \omega_{\varepsilon_1=0}^2}. \quad (\text{R1})$$

For lp-SNO, the conventional plasmon energy (taking from the energy where the loss function is maximum) is ~ 1.90 eV, while the zero-crossing of $\varepsilon_1(\omega)$ happens at ~ 1.84 eV.

Using the above equation, we get $\frac{\hbar}{\tau} = \sim 0.47$ eV. In the second method, the free electron scattering can be estimated from the full-width half-maximum (FWHM), $2\hbar\Gamma$, of the conventional plasmon peak taken from the $-\text{Im}[\varepsilon^{-1}(\omega)]$ spectrum. For lp-SNO, this value is ~ 0.45 eV, which is consistent with the value obtained from the blue-shift method.

The free electron scattering contributes to the dephasing of the conventional plasmon, which means that the conventional plasmon dephasing time, T_2 , can be obtained using

$$T_2 = \frac{1}{\Gamma} = 2\tau. \quad (\text{R2})$$

Thus, a free electron scattering of ~ 0.47 eV is equivalent to a conventional plasmon dephasing time of ~ 2.8 fs in lp-SNO. As for mp-SNO, its conventional plasmon energy and $\varepsilon_1(\omega)$ zero-crossing is ~ 1.70 eV and ~ 1.62 eV, respectively, which means that its free electron scattering and conventional plasmon dephasing time is ~ 0.52 eV and ~ 2.5 fs, respectively.

On the other hand, the microscopic theory of correlated plasmons in strongly-correlated systems is much less established compared to that of conventional plasmons. While as a first approximation we can use the FWHM of the correlated plasmon peaks in hp-SNO to estimate the dephasing time of the correlated plasmons, the scattering and dephasing mechanisms in correlated systems might be different from those in weakly-correlated metals due to, for instance, long-range correlation effects, renormalized effective mass and redistributed charge

arrangement. The FWHM of the C_1' , C_2' , and C_3' peaks of hp-SNO is ~ 0.7 eV, ~ 0.68 eV, and ~ 0.38 eV, respectively. (Note that we cannot use the blue-shift method because hp-SNO has no $\epsilon_1(\omega)$ zero crossing.) By following the same rules as that of conventional plasmons, a correlated plasmon dephasing time is ~ 1.9 fs, ~ 1.9 fs, and ~ 3.5 fs, respectively for the three correlated plasmon energies.

We have added this discussion about the dephasing of conventional and correlated plasmons in the revised manuscript.

Comment R1.3: B. Originality and interest: if not novel, please give references

Potentially novel, but not enough evidence.

Response R1.3: To our knowledge, our result is the first observation of correlated plasmons in correlated electron systems. This is also mentioned by Reviewer #3.

Comment R1.3:

C. Data & methodology: validity of approach, quality of data, quality of presentation

Experiments connected to exciting plasmons

(I) grating, Otto/Kretschmann configuration)

Response R1.3: We note that the grating, Otto, and Kretschmann configurations are several established techniques that are used to excite conventional surface plasmon resonance (SPR) using photons. Surface plasmons themselves are plasmons that exist at the metal-dielectric (usually ambient air) interfaces due to the abundance of free electrons at the metal surface. However, despite their ubiquitous use in the study and applications of conventional SPR,

based on the following reasons such configurations might not be directly suitable to study the correlated plasmons found in the insulating hp-SNO sample.

1. The hp-SNO sample is an insulator with a Mott-like behavior, as seen from optical conductivity (Fig. 1), spectral weight transfer signatures (Fig. 2) and its transport results (paragraph 4). This means that its surface does not have free electrons that can interact with photons to form surface plasmons. From this reason alone, we can already see that the grating, Otto, and Kretschmann configurations might not be suitable to study surface plasmons in insulating hp-SNO because the sample has no (conventional) surface plasmon to begin with.
2. Based on thickness-dependent studies (see Fig. R1 and Fig. 4 of the revised manuscript as well as the discussion below), we can also infer that the correlated plasmons that we observed in hp-SNO are most likely bulk plasmons, as they do not occur in the very thin film sample. Furthermore, as discussed in our manuscript, their mode of excitation involves electron confinement by extra oxygen planes, which are embedded in the bulk crystal structure of the sample. This also limits the applications of the grating, Otto, and Kretschmann configurations in studying of the (bulk) correlated plasmons of hp-SNO since their function is to excite surface plasmons rather than bulk plasmons.
3. Furthermore, the grating, Otto, and Kretschmann configurations are needed to excite SPR because free-space photons have different dispersion compared to surface plasmons: for a given frequency, free-space photons always have smaller wave vectors compared to surface plasmons. Thus, in conventional SPR these external configurations are needed to increase the wave vector of the photons before they can couple with surface plasmons at

the metal surface. For the same reason, surface plasmons cannot directly decay radiatively into outgoing photon because of this difference in dispersions.

On the other hand, the $\varepsilon_2(\omega)$ and loss function of hp-SNO (Figure 1d) shows that the correlated plasmon peaks appear at *both* the loss function and $\varepsilon_2(\omega)$ of hp-SNO, showing that these correlated plasmons *can already directly couple* with free-space photons without any special external configurations.

Before we elaborate further, we would like to remind that the transverse $\varepsilon_2(\omega)$ (or ε'' as usually used in the plasmonic community) measures the elastic dielectric response of a material. In this elastic process, an incoming photon with a certain energy can couple with an excitation if the excitation energy is matched with the incoming photon. If the excitation can decay radiatively into an outgoing photon with the same exact energy as the incoming photon, the outgoing photon will be detected as a peak in the $\varepsilon_2(\omega)$ spectra. If the excitation cannot be excited by or decay into a photon or if there is no energy-matched excitation to couple with, no peak will be detected in the $\varepsilon_2(\omega)$ spectra.

Meanwhile, peaks in the longitudinal loss function spectra, $-\text{Im}[\varepsilon^{-1}(\omega)]$, indicate the existence of plasmons in the material. For sub-X-ray photons, the photon momentum transfer, q , is finite but approaches zero because it is much less than the crystal momentum. In this limit, the distinction between longitudinal (l) and transverse (t) $\varepsilon(\omega)$ vanishes, *i.e.*, $\lim_{q \rightarrow 0} \varepsilon_l(q, \omega) = \varepsilon_t(q, \omega)$, which allows sub-X-ray optical spectroscopy to

also probe plasmonic properties of the material in the low momentum transfer limit (see Ref. 19 for details).

For the metallic lp-SNO and mp-SNO, despite the huge conventional plasmon peaks at ~ 1.9 eV and ~ 1.7 eV, respectively, in their loss function spectra, their corresponding $\epsilon_2(\omega)$ spectra show no significant peak at these energies (Figs. 1b,c). This shows that the conventional plasmons indeed cannot directly couple with free-space photons, as mentioned above.

Meanwhile, in insulating hp-SNO the correlated plasmon peaks not only appear in the loss function spectra *but also* in the ϵ_2 spectra of hp-SNO (C_1 , C_2 , and C_3 peaks in Fig. 1d). This shows that these correlated plasmons *can already* be excited by, decay radiatively into, and thus directly couple with free-space photons without any special external configurations such as grating, Otto, and Kretschmann configurations. We hope that this paper can be a basis for future experimental and theoretical efforts to study this new kind of plasmonic excitation.

Therefore, in our understanding while the grating, Otto, and Kretschmann configurations may be suitable to excite conventional surface plasmons, such configurations may be challenging for correlated plasmons. It is important to note that our current analysis and conclusion do not depend on these kind of configurations (please also see our responses below).

To clarify this point, we have added a new discussion about this plasmon-photon coupling in paragraph 10 of the revised manuscript where we discuss the differences between correlated plasmons and conventional plasmons.

Comment R1.4: (II) Plasmonic size dependence

Response R1.4: As Reviewer #1 requested, we have grown different thicknesses of hp-SNO from ~20 nm to ~81 nm. For comparison, the standard hp-SNO sample presented in Fig. 1 has a thickness of ~168 nm. The resulting complex dielectric function, $\varepsilon(\omega) = \varepsilon_1(\omega) + i\varepsilon_2(\omega)$, and loss function, $-\text{Im}[\varepsilon^{-1}(\omega)]$, of the samples are shown in Fig. R1 and Fig. 4 of the revised manuscript. Interestingly, both the energy and number of excited correlated plasmons change when the film thickness changes. We find that as the thickness decreases, the number of correlated plasmon peaks decreases. When the film thickness decreases further to 20 nm, the correlated plasmon disappears. This means that besides using the deposition pressure, the correlated plasmons can also be tuned by varying the thickness of the film. This is again fundamentally different from conventional plasmons where only the plasmon energy is changing when the metal nanoparticle size is changed.

The disappearance of correlated plasmons in the 20 nm film is particularly interesting. To investigate this further, we study the optical conductivity and spectral weight transfer of the thickness-dependent films shown in Fig. R2 and Fig. 5 of the revised manuscript. It can be seen that as the film thickness decreases, the spectral weight of the first interband region (> 4.3 eV), W_3 , increases. This means that the thinner films have less wide-range spectral weight transfer that signifies their correlation strength. In particular, the W_3 of the 20 nm hp-SNO film is almost as high as that of the metallic mp-SNO (see Figs. 2b,5b), which means that the 20 nm hp-SNO is most likely a weakly-correlated conventional band insulator instead of a strongly-correlated Mott-like insulator like the other, thicker hp-SNO films. This reinforces the connection between the correlated plasmons and electronic correlation: as electronic

correlation becomes weaker (whether in low-pressure metallic films or thin conventionally-insulating films), the correlated plasmons become weaker and ultimately vanish.

We have added these new data and discussion about thickness-dependent hp-SNO into the revised manuscript.

Figure R1. Complex dielectric function and loss function spectra of hp-SNO films with varying thicknesses. a, Real part of complex dielectric function, $\epsilon_1(\omega)$, of hp-SNO films

with varying thicknesses. **b,** Imaginary part of complex dielectric function, $\epsilon_2(\omega)$, and loss

function, $-\text{Im}[\epsilon^{-1}(\omega)]$, of ~ 81 nm hp-SNO film. Here, two correlated plasmon peaks are

observed at ~ 3.2 eV and ~ 4.4 eV. **c,** The $\epsilon_2(\omega)$ and $-\text{Im}[\epsilon^{-1}(\omega)]$ of ~ 52 nm hp-SNO film.

Here, the only correlated plasmon peak is observed at ~ 4.0 eV. **d,** The $\epsilon_2(\omega)$ and

$-\text{Im}[\epsilon^{-1}(\omega)]$ of ~ 20 nm hp-SNO film. No correlated plasmon peak is observed in this

sample. For reference, the standard hp-SNO film which has three correlated plasmon peaks

has a thickness of ~ 168 nm.

Figure R2. Optical conductivity spectra, $\sigma_1(\omega)$, and spectral weight, W , of hp-SNO

films with varying thicknesses. a, The $\sigma_1(\omega)$ of hp-SNO films with thicknesses of ~81 nm, ~52 nm, and ~20 nm. **b,** Evolution of W of three energy regions: W_1 (0.6-2.0 eV), W_2 (2.0-4.3

eV), and W_3 (4.3-6.5 eV) across hp-SNO films with varying thicknesses. The W_{tot} is the total W from 0.6 to 6.5 eV. The W_3 of mp-SNO (purple filled circle) is also shown for comparison. The abbreviation ‘CP’ stands for ‘correlated plasmon’.

Comment R1.5: (III) Transmission/Reflection/Absorption measurement of the films

Response R1.5: We would like to note that spectroscopic ellipsometry already uses specular reflection (and sometimes transmission) geometry in its experimental setup (see Ref. 28 of the revised manuscript for more details). In this particular paper, we use spectroscopic ellipsometry with a specular reflection geometry setup. The advantage that spectroscopic ellipsometry has compared to conventional reflection/transmission measurements is its precise incoming and outgoing light polarizations control and detection. Because of this, spectroscopic ellipsometry allows us to extract the complex dielectric function ($\varepsilon(\omega) = \varepsilon_1(\omega) + i\varepsilon_2(\omega)$) of the material with superior precision compared to standard unpolarized reflection/transmission measurements. Since $\varepsilon(\omega)$ is the most fundamental optical response of a material, from $\varepsilon(\omega)$ we can further extract the refractive index (n), extinction coefficient (κ), absorption coefficient (α), and reflectivity of the materials via the Fresnel equations (see Eqs. 1-4 of the revised manuscript). To demonstrate this, in Fig. R3 and Supplementary Fig. 7 of the revised manuscript we show the n , κ , normal-incident reflectivity, and absorption coefficient of the lp-SNO, mp-SNO, and (168 nm) hp-SNO films. Meanwhile in Fig. R4 and Supplementary Fig. 12 of the revised manuscript we show the n , κ , normal-incident reflectivity, and absorption coefficient of the thickness-dependent hp-SNO films. All of these figures are extracted from spectroscopic ellipsometry data.

In particular, from the normal-incident reflectivity spectra of lp-SNO, mp-SNO, and (168 nm) hp-SNO (Figs. R3 and Supplementary Fig. 7 of the revised manuscript), it can be seen that the reflectivity minimum of lp-SNO and mp-SNO occurs at ~ 2.1 eV and 1.9 eV, respectively, close to their respective conventional plasmon energies of ~ 1.9 eV and ~ 1.7 eV. Meanwhile, hp-SNO has multiple reflectivity minima at ~ 1.9 eV, ~ 3.2 eV, and ~ 4.2 eV, which are also very close to its correlated plasmon energies of ~ 1.7 eV, ~ 3.0 eV, and ~ 4.0 eV, respectively.

Figure R3. Refractive index, extinction coefficient, normal-incident reflectivity, and absorption coefficient of lp-SNO, mp-SNO, and hp-SNO. **a**, Refractive index, n , of lp-SNO, mp-SNO, and hp-SNO. **b**, Extinction coefficient, κ , of lp-SNO, mp-SNO, and hp-SNO. **c**, Normal-incident reflectivity of lp-SNO, mp-SNO, and hp-SNO. **d**, Absorption coefficient, α , of lp-SNO, mp-SNO, and hp-SNO. The legend is shown in **a**. Insets show parts of the spectra zoomed out to show the full extent of the extinction coefficient and normal-incident reflectivity of lp-SNO and mp-SNO.

Figure R4. Refractive index, extinction coefficient, normal-incident reflectivity, and absorption coefficient of hp-SNO films with varying thicknesses. a, Refractive index, n , of hp-SNO films with varying thicknesses. **b,** Extinction coefficient, κ , of hp-SNO films with varying thicknesses. **c,** Normal-incident reflectivity of hp-SNO films with varying thicknesses. **d,** Absorption coefficient, α , of hp-SNO films with varying thicknesses. The legend is shown in **a**. The n , κ , normal-incident reflectivity, and α of the standard ~ 168 nm hp-SNO are also shown for reference.

Comment R1.6: (IV) EELS experiment are all lacking.

Response R1.6: We have tried to measured X-ray spectroscopy spectroscopy on these films, however we were facing charging effects due to the insulating nature of the correlated plasmon samples. Therefore, EELS measurement is not an appropriate measurement to study insulating samples such as hp-SNO. To elaborate, EELS measurement uses high-energy electrons to probe the loss function of a material. This technique is suitable and often used to measure the plasmonic properties of metals because metals are conductors. However, hp-

SNO is a Mott-like insulator, and the injection of hot electrons into the sample introduced significant charging effects that would make the resulting EELS spectrum unreliable. This is the reason why we chose to investigate the plasmonic properties of hp-SNO using optical techniques such as spectroscopic ellipsometry rather than EELS, in which there is no charging effects in the “photon-in photon-out” measurement. Furthermore, in the low momentum transfer regime the complex dielectric function and loss function obtained using EELS converge to the one measured using optics (see above and Ref. 19 for details).

Comment R1.7: D. Appropriate use of statistics and treatment of uncertainties

Not enough samples, and systematic studies varying the film thicknesses or interlayers.

Response R1.7: As shown above, we have done extensive study on thickness-dependent films due to Reviewer #1’s request.

Comment R1.8: G. Not enough credit/references that combine the work related to plasmonic and Mott-effect.

Response R1.8: The result that we present in this manuscript is really the first successful experimental attempt at studying plasmons in strongly-correlated materials such as Mott insulators. This is also mentioned by Reviewer #3. To our knowledge, Ref. 7 (published very recently on December 2014) that we have cited was also the first (and so far perhaps only) theoretical effort that directly study the effects of strong electronic correlation and long-range Coulomb interactions on the properties of plasmons. If Reviewer #1 knows any other works that combine both plasmonic and Mott effects, we are open for suggestions.

Reviewer #2 (Remarks to the Author):

Comment R2.1: In the reviewed manuscript, the authors present an extensive and conclusive study of a new type of tunable correlated plasmons observed in a wide bandgap, Mott-like insulating oxide from the $\text{Sr}_{1-x}\text{Nb}_{1-y}\text{O}_{3+\delta}$ family. Due to the long-range Coulomb interactions, the electrons in the oxide are strongly correlated and the corresponding collective excitations of the correlated electrons result in multiple plasmon frequencies with low losses. The authors experimentally observed the correlated plasmons and bring us to a new plasmonic material in the visible-ultraviolet range, based on wide bandgap and strongly correlated materials, tackling one of the most challenging issues in plasmonics research. The authors interpret the physical origin of the observed tunable correlated plasmons in $\text{Sr}_{1-x}\text{Nb}_{1-y}\text{O}_{3+\delta}$ as the nanometer-spaced electronic confinement of extra oxygen planes resulting from crystal growth under high oxygen pressure, which is fully supported by the density functional theory calculations together with a coupled oscillator model.

Response R2.1: First of all, we would like to thank Reviewer #2 for his/her support and feedback for our manuscript.

Comment R2.2: I have only a few minor questions/comments for the authors to address before accepting this work for publication in Nature Communications.

(1) SNO films with different concentration of oxygen are produced and their crystallinity is characterized by TEM and powder X-rays. Is there any direct indication of the stoichiometric ratio for oxygen in SNO films?

Response R2.2: We admit that at this stage, it is still very challenging to quantify precisely the oxygen (O) content in films, particularly in oxide films on oxide substrates such as the lp-SNO, mp-SNO, and hp-SNO films that are deposited on LaAlO₃ substrates. In addition to characterizations presented in the manuscript, we did try to measure using other spectroscopic methods, including x-ray photoemission spectroscopy. However, because of the insulating nature of the films, we were facing charging effects and the data were not reliable.

Nevertheless, as discussed in Response R2.4 below and in Ref. 12, SrNbO_{3+δ} with different oxygen content (*i.e.*, different δ , with $\delta \geq 0$) has different extra oxygen plane spacing and repetition pattern. Thus, in this case the oxygen content of the lp-SNO, mp-SNO, and hp-SNO films can be estimated by considering the local oxygen defect microstructures of the films. From the transmission electron microscopy (TEM) image in Fig. 3e, it can be seen that the extra oxygen planes structure of hp-SNO matches with that of bulk SrNbO_{3.4}, which means that hp-SNO should have an oxygen content close to that of SrNbO_{3.4}. On the other hand, Fig. 3a shows that lp-SNO has almost no extra oxygen planes, which means that it should have an oxygen content close to that of SrNbO₃. Meanwhile, since mp-SNO is deposited under an oxygen pressure between those of lp-SNO and hp-SNO, it should also have an oxygen content between those of lp-SNO and hp-SNO. The density of the oxygen planes has also been measured using selected-area electron diffraction (SAED) mode of TEM (Supplementary Fig. 8), where it is shown that lp-SNO has almost no extra oxygen planes while the density of extra oxygen planes in hp-SNO is four times higher than in mp-SNO. We then calculated electron density (see **Figure R6 below**) and find that there is a good match between the electron density calculations and the experimental data. We have added this discussion into the Supplementary Discussion of the revised manuscript.

Comment R2.3: During growth, the crystal defects could be produced by the exchanged cations, such as vacancies. Can the authors estimate the effect of the crystal defects on the correlated plasmons?

Response R2.3: From Rutherford Back Scattering (RBS), we estimate that the films have an Sr/Nb cationic ratio of 0.94 (see Methods), which means that the films have 6% Sr vacancies in their composition. We do not detect the presence of other cationic vacancies within our measurement resolution. At this stage the effects of these Sr vacancies on the properties of the correlated plasmons are still unclear, and this requires a detailed systematic study with series of samples involving various cationic vacancies, which is beyond our current scope. We note that in our current analysis, we attribute the origin of the correlated plasmons to the strong electronic correlations induced by the electronic confinement caused by the presence of the extra oxygen planes. So far, this analysis seems to be adequate to explain our results.

Comment R2.4: (2) The authors should add the description of the growth mechanism for oxygen planes formation in hp-SNO films. Why can the oxygen atoms form pure planes without forming other atomic arrangement manners?

Response R2.4: In short, the oxygen planes are formed via self-assembly. To elaborate, we would like to clarify that the extra oxygen planes are not the only structural changes that happen when the O concentration is increased. Structurally, the $\text{SrNbO}_{3+\delta}$ family of oxides is a member of a larger group of oxides with a general formula $A_nB_nO_{3n+2}$. The crystal structures of the compound with $n = 4.5, 5, 6,$ and ∞ are shown in Fig. R5 taken from Ref. 12. This structural group of oxides is also sometimes referred to as ‘layered perovskites’, since their crystal structure is that of perovskite ‘supercells’ that are layered against each other

every few unit cells along one particular direction (in this case [110]). Nearest-neighbor perovskite supercells have a height difference (along the [011] direction) of a half unit cell, while next-nearest-neighbor supercells have the same height. At the end of each supercell, there are half per unit cell oxygen ions that are not compensated by cations. Thus, the uncompensated oxygen ions from two neighboring supercells would form an extra oxygen plane along the [101] direction, as shown in our TEM image in Fig. 3. The subscript n indicates the size of each perovskite supercell and thus the repetition of this pattern. In particular, for $\text{SrNbO}_{3+\delta}$ and $n = 5$, we would get the structure of $\text{SrNbO}_{3.4}$, while for $n = \infty$ we would get the (unlayered) perovskite structure of SrNbO_3 with no extra oxygen planes. These atomic arrangements have been directly observed using TEM in our paper as well as in previous studies such as Refs. 11 and 12. Furthermore, in the DFT calculations (Fig. 4b), we have also taken this atomic arrangement into account in calculating the electron density and relaxed the superlattice used in the calculations accordingly (see Methods). These experimental and theoretical findings show that these layered perovskite structures with extra oxygen planes are indeed the lowest-energy self-assembled atomic structure of $\text{SrNbO}_{3+\delta}$. We have added this discussion to the Supplementary Discussion of the revised manuscript.

Fig. 3. Sketch of the idealized (i.e. non-distorted) crystal structure of the $n=4.5, 5, 6$ and ∞ members of the perovskite-related layered homologous series $A_nB_nO_{3n+2}$ projected along the a -axis. The stoichiometries are also given as ABO_x with its corresponding ideal oxygen content $X=3+2/n$. Circles represent the A cations. Within the layers the corner-shared BO_6 octahedra extend zig-zag-like along the b -axis and chain-like along the a -axis (see also Fig. 4). The layers are n octahedra thick, thus the thickness of the layers rises with increasing n . For $n=\infty$ the three-dimensional perovskite structure ABO_3 is realized. The $n=4.5$ member represents the well-ordered stacking sequence $n=5, 4, 5, 4, \dots$. Light and heavy drawing of the BO_6 octahedra indicates a height difference perpendicular to the drawing plane of about 2 \AA , the half of the octahedron body diagonal and B -O bond length. Filled and open circles indicate A cations also differing in height by this distance perpendicular to the drawing plane.

Figure R5. The atomic structure of $A_nB_nO_{3n+2}$ structural group, which $SrNbO_{3+\delta}$ family of oxides are part of. The figure and associated caption is taken directly from Ref. 12.

Comment R2.5: (3) Although the oxygen planes can be clearly observed from the TEM images of hp-SNO, their global arrangement and detailed structures in the hp-SNO films remain unclear. Since the exact arrangement of oxygen planes is critical for the density functional theory calculations, the authors have to describe this aspect in great detail and specify the dimensionality of the space confinement on the carriers.

Response R2.5: The global arrangement of the atomic structure of hp-SNO is shown Fig. 3c. It can be seen that the extra oxygen planes have two orientations: along [101] and [-101] directions, which are perpendicular to each other (see also the discussion in Supplementary Discussion). This causes twin domains to be formed in the film structure based on the direction of the oxygen planes: on some domains, the oxygen planes are along the [101] direction, while on others they are along the [-101] direction. From the scale bar (8 nm), it can be estimated that the width of one of these ‘orientation domains’ are around 24 nm, or ~60 unit cells. If we were to model these domains using DFT, we would have to calculate using a supercell of at least 60 unit cells (which contain more than 1000 atoms) or possibly even more. This is a task that is beyond current computational capabilities. We note that our current analysis and conclusion do not depend on this.

Furthermore, as also mentioned by Reviewer #3, DFT itself has a limitation in calculating correlated electron systems because it cannot take into account correlation effects properly, in particular the spectral density transfers relevant for Mott physics (see, for instance, Ref. 7). Thus, one should not take the DFT results at face value at this early stage. Our intention for the theoretical parts is to provide a qualitative, phenomenological explaining this new form of correlated plasmons. To clarify this point, we have added a new discussion on the limitations of DFT calculations in the revised manuscript.

As for the dimensionality of the space confinement, since for the $\text{SrNbO}_{3.4}$ structure the extra oxygen planes occur every 5 unit cells and the lattice constant of the film is 4.04 \AA , this means the electrons are confined within a space of $\sim 2 \text{ nm}$. We have added this information in the revised manuscript.

Comment R2.6: (4) The mode of correlated plasmons is dependent on the spaced confinement that can be tuned by changing the density of oxygen planes. I suggest the authors carrying out additional DFT calculations for a series of oxygen plane densities in hp-SNO films. This may help understand the exact role of oxygen planes in generating the observed tunable and correlated plasmons.

Response R2.6: In addition to SrNbO_3 and $\text{SrNbO}_{3.4}$ (Figs. 6a,b), as requested by the referee, we perform density functional theory (DFT) calculations on $\text{SrNbO}_{3.33}$ and $\text{SrNbO}_{3.5}$ (Fig. R6 and Supplementary Fig. 12 of the revised manuscript). From these calculations, the effect of different oxygen concentration in $\text{SrNbO}_{3+\delta}$ is to change the spacing (and thus density) of the extra oxygen planes. Thus, if the extra oxygen planed density is changed, the DFT calculations shows that the electronic confinement between the oxygen planes is changed. Again, it should be noted that DFT calculations are not adequate explain the spectral weight transfer observed in Fig. 2 and the correlated plasmons themselves, as this can only be explained via optical spectrum and electronic correlations (please also see, for instance, Ref. 7).

Figure R6. Theoretical Nb-4d electron density iso-surface of SrNbO₃, SrNbO_{3.33}, SrNbO_{3.4}, and SrNbO_{3.5}. **a**, Theoretical Nb-4d electron density iso-surface of SrNbO₃ superimposed on SrNbO₃ crystal structure (Sr = green, Nb = blue, O = red, iso-surface = yellow). **b**, Theoretical Nb-4d electron density iso-surface of SrNbO_{3.33} superimposed on SrNbO_{3.33} crystal structure. **c**, Theoretical Nb-4d electron density iso-surface of SrNbO_{3.4} superimposed on SrNbO_{3.4} crystal structure. **d**, Theoretical Nb-4d electron density iso-surface of SrNbO_{3.5} superimposed on SrNbO_{3.5} crystal structure. Black dashed lines denote the oxygen walls.

For SrNbO_{3.33} (Fig. R6a and Supplementary Fig. 12a), the extra oxygen spacing is 6. For SrNbO₃, there is no extra oxygen spacing. Along this trend, one may say a decrease in the electron confinement and, by extension, a decrease in the electronic correlation. This decrease in the electronic correlation would diminish the correlated plasmons, as seen in the

$\varepsilon(\omega)$ of mp-SNO (which is estimated to have a lower oxygen concentration compared to hp-SNO) where there is only one small correlated plasmon peak at ~ 3.5 eV (B_2' peak in Fig. 1c). On the other hand, if the oxygen concentration is increased to $\text{SrNbO}_{3.5}$ (Fig. R6b and Supplementary Fig. 12b), the extra oxygen spacing would decrease to 4 unit cells. However, this increase of extra oxygens is enough to deplete the confined electrons, and as a result $\text{SrNbO}_{3.5}$ is found to be an (ferroelectric) insulator (see Refs. 17 and 18).

Another remark from the DFT calculations is that we also find different orbital configurations for each cases, which may be related to the different types of plasmons that are excitable in different films: conventional plasmons in SrNbO_3 , correlated plasmons in $\text{SrNbO}_{3.4}$ and a mixture of conventional and correlated plasmon in $\text{SrNbO}_{3.33}$. Since the DFT calculations cannot clarify the existence of correlated plasmons, at this stage it remains to be seen how to quantitatively link between these different orbital configurations with various types of plasmons.

Comment R2.7: (5) From Fig. 1a-c, the lp-SNO and mp-SNO films show the large peaks of $A'1$ and $B'1$, respectively. These peaks coincide with the zero-crossing of $\varepsilon_1(\omega)$, indicating that the peak comes from a conventional plasmon resonance. However, both peaks $A'1$ and $B'1$, have a small blue-shift compared to the zero-crossing. This blue-shift is not discussed in the manuscript and is important for the conventional plasmon resonance.

Response R2.7: We would like to thank Reviewer #2 for bringing up this issue that we did not explicitly discuss it in the previous version of our manuscript. In short, the blue shift is due to the free electron scattering and thus the dephasing of conventional plasmons. To elaborate, from Supplementary Eq. 6b, the $\varepsilon_1(\omega)$ based on the Drude model is

$$\varepsilon_1(\omega) = \varepsilon_1(\infty) - \frac{nme^2\omega^2}{(m\omega^2)^2 + \left(\frac{m}{\tau}\omega\right)^2} = \varepsilon_1(\infty) \left[1 - \frac{\omega_p^2}{\omega^2 + \left(\frac{1}{\tau}\right)^2} \right]. \quad (6b)$$

At the energy where $\varepsilon_1(\omega)$ goes to zero (*i.e.*, $\omega_{\varepsilon_1=0}$) the equation becomes

$$\omega_p = \sqrt{\omega_{\varepsilon_1=0}^2 + \left(\frac{1}{\tau}\right)^2}. \quad (R3)$$

Thus, it can be seen that the free electron scattering would always cause the conventional plasmon energy to be blue-shifted compared to the zero-crossing of $\varepsilon_1(\omega)$. Only when there is no or negligible electron scattering would the conventional plasmon energy would be exactly equal to the zero-crossing of $\varepsilon_1(\omega)$.

Conversely, using Eq. R3 we can also extract the free electron scattering from $\varepsilon_1(\omega)$ and the loss function spectra. For example, the conventional plasmon energy of lp-SNO (taken from the energy where the loss function becomes maximum) is ~ 1.90 eV, while the zero-crossing of $\varepsilon_1(\omega)$ happens in lp-SNO at ~ 1.84 eV. Using Eq. R3, we get $\frac{\hbar}{\tau} = \sim 0.47$ eV. This value is also consistent with the full-width half-maximum (FWHM), $2\hbar\Gamma$, of the conventional plasmon peak of lp-SNO, which is ~ 0.45 eV. This free electron screening would contribute to the dephasing of the conventional plasmon, and the conventional plasmon dephasing time, T_2 , can be obtained from

$$T_2 = \frac{1}{\Gamma} = 2\tau. \quad (R3)$$

For lp-SNO, a free electron scattering of ~ 0.47 eV translates to a conventional plasmon dephasing time of ~ 2.8 fs. The same treatment can also be used to calculate the conventional plasmon dephasing time of mp-SNO. With a plasmon energy of ~ 1.70 eV and $\varepsilon_1(\omega)$ zero-

crossing of ~ 1.62 eV, we get a free electron scattering of ~ 0.52 eV and a conventional plasmon dephasing time of ~ 2.5 fs for mp-SNO. We have added this discussion into the revised manuscript.

Reviewer #3 (Remarks to the Author):

Comment R3.1: The paper is very interesting; indeed, to my knowledge there were no successful attempts to study experimentally plasmons in strongly correlated systems. As a first attempt of this kind, the work deserves to be published.

Response R3.1: First of all, we would like to thank Reviewer #3 for his/her support for our manuscript.

Comment R3.2: Theoretical part is a bit naive; at the same time, it is known that in the limit of small wave vectors the classical model of coupled oscillators may be qualitatively correct. Of course, it cannot be sufficient to describe EELS with finite wave vector, and this should be said explicitly.

Response R3.2: Yes, we understand this problem and agree with Reviewer #3. In the revised manuscript we have clarified that our results are obtained within the limit of low wave vectors.

Comment R3.3: At the same time, it would be nice to have a more detailed comparison with theoretical results (Ref.7); the most important in the theory is not that it can be a renormalization of parameters like effective mass etc. but that in principle several types of plasmon excitations are possible.

Response R3.3: Yes, we agree that our most important result is that there are several types of plasmons that can be excited and the correlated plasmons may be excited on several

frequencies (or energies). In metals the conventional plasmons dominate while in correlated systems correlated plasmons, which have fundamentally different properties and origins from conventional plasmons, are excitable instead. We have further emphasized this in the revised manuscript.

As for a more detailed comparison with Ref. 7, we agree that to some extent, our results do share a resemblance with the theoretical results of Ref. 7. In Ref. 7, strong correlations and long-range Coulomb interactions can result in multiple plasmon energies. Interestingly, these plasmon energies appear to coincide with the multiples of effective correlation strength, U^* , *e.g.*, at U^* and $U^*/2$. This is similar to the correlated plasmons that we observe. The correlated plasmons have multiple plasmon energies, and these plasmon energies happen to have a seemingly ordered ratio (1.7, 3, and 4). However, this ratio is not as straightforward as the U^* and $U^*/2$, in which perhaps some other mechanisms may also play important role. We have added this discussion in the revised manuscript.

Comment R3.4: It is not completely clear how the results of density-functional calculations (which do not take into account properly correlation effects, in particular, spectral density transfer relevant for Mott physics) are related to the experimental results. Currently, much better calculations can be done within, *e.g.*, LDA+DMFT (dynamical mean field theory). I do not mean that this kind of calculations is obligatory for this particular paper, this is first of all an experimental work with some auxiliary theoretical material, but, at least, all restrictions of the method used in the calculations should be clearly formulated. In general, I believe this is an interesting and important work of a broad interest which may be suitable for publication in Nature Communications but I would recommend the authors to improve discussion to make theoretical part and its relevance for experiment more clear.

Response R3.4: Yes, we agree that present calculations do have limitations and we have added a discussion about this limitation in the revised manuscript. We hope in the future a more rigorous theoretical treatment, such as or GW-BSE or LDA+DMFT with proper vertex corrections suggested by Reviewer #3, can shed a more complete picture on the mechanism of this correlated plasmon phenomenon.

Reviewer #1 (Remarks to the Author):

Although I think the idea of correlated plasmonic is a potentially interesting one, the lack of clear direct evidence of correlation and more importantly how this is a plasmonic effect is now more striking with the size-dependent thin-film measurement.

First of all, we would like to thank Reviewer #1 for his/her questions and comments. Below, we provide point-by-point answer to all of Reviewer #1's questions.

We would like to reiterate that it is well established that anomalous spectral weight transfer over a wide energy range is the most direct evidence of strong correlations. Such spectral weight transfer can be measured precisely through complex dielectric response (or optical conductivity) over a broad energy range because it fulfils the f -sum rule. The fundamental physical reasons of why this is the case have been discussed in much details in a number of papers such as Refs. [11, 24, 30-35] that we have cited in the revised manuscript. This method has been used to determine the strong correlations in strongly-correlated materials, such as cuprates (Refs. [11, 30, 33]), vanadates (Ref. [34]), and manganites (Refs. [11, 30, 35]). We therefore believe that we have provided a clear direct evidence of correlations in this system (please see the discussion below for details).

As an example, Basov's group mentioned by Reviewer #1 also used this anomalous wide-range spectral weight signature to examine the strong correlations in VO_2 and V_2O_3 (*Phys. Rev. B* **77**, 115121 (2008), cited in the revised manuscript as Ref. [34]). VO_2 is a strongly-correlated material which undergoes a temperature-induced metal-insulator transition (MIT)

at $T = 340$ K from a low-temperature Mott-insulating phase to a high-temperature rutile metallic phase. In this paper, Basov's group used ellipsometry (the same technique that we also use in this manuscript) to study the spectral weight (SW) of VO_2 (and V_2O_3) below and above the Mott transition temperature. They found that there was a large change in SW across wide energy range (above 6 eV) when the MIT happened at 340 K. Regarding this large spectral weight change, we quote verbatim from this Basov's paper (emphasis is ours) that:

“The rather large energy scale over which changes in $\text{SW}(\omega)$ are seen across the metal-insulator transition ought to be taken as **direct evidence for the predominance of correlation effects**, and hence an indication of a Mott transition. Similar large energy scales are involved in spectral weight changes in doped Mott insulators, for example, in the cuprates.”

As we have discussed extensively in our manuscript (paragraphs 9 and 10 of the Results section and Fig. 2), at the onset of the MIT between the metallic mp-SNO and insulating hp-SNO, we also found these anomalous spectral weight transfers over a wide energy range, similar to what Basov's group (and other papers such as Refs. [11, 24, 30-35] that we cited) found during the MIT between metallic and insulating phases of VO_2 (and other correlated materials). First, the spectral weight changes between mp-SNO and hp-SNO are anomalous because it does not conform to what conventional band theory (which does not include strong correlation effects) predicts (see paragraph 10 of the Results section). Second, the spectral weight transfer happens over a wide range of energy because the total spectral weight decreases substantially below 6.5 eV (Fig. 2). Since spectral weight obeys the f -sum rule (paragraph 9 of the Results section), this decrease ought to be compensated by a spectral weight increase above 6.5 eV, thus implicating spectral weight transfers over more than 6.5

eV in energy scale. For comparison, the large spectral weight changes in the MIT between the metallic and insulating phases of VO₂ also happened over an energy scale of 6 eV (Ref. 34), similar to what we observe in the MIT between mp-SNO and hp-SNO.

From the above discussion, it is clear that there is indeed an anomalous spectral weight transfer over a wide energy range in the MIT between mp-SNO and hp-SNO, and that this anomalous spectral weight transfer is indeed a *direct evidence* of strong correlation effects at the MIT between metallic mp-SNO and insulating-like hp-SNO. Thus, we would like to again state that there is indeed a clear evidence of strong correlations in hp-SNO, and that hp-SNO is most likely a Mott-like insulator.

This is clearly not suitable for Nature Communications.

- Also, other groups have discussed correlations in regard to such Mott-type or exotic materials, such as the group by Basov or Haglund (VO₂), Zheludev (YBCO superconducting plasmonics) and Cavalleri (Cuprate superconductors for transient plasmonic response) to name a few.

We agree that these groups had indeed tried to investigate the plasmonic properties of strongly-correlated materials. However, as explained below, the nature of plasmons in their studies is that of typical conventional plasmons and is different from the correlated plasmons we observed in our hp-SNO film. For examples, Basov's and Haglund's groups are investigating the plasmonic properties of VO₂ (*Appl. Phys. Lett.* **105**, 041117 (2014), *Nano Lett.*, **11**, 1025 (2011), and *Opt. Express* **23**, 6878 (2015), and cited as Refs. [6-8] in the revised manuscript, respectively), while Zheludev's and Cavalleri's groups are studying the

plasmonic properties of cuprates (*Appl. Phys. Lett.* **97**, 111106 (2010) and *Nat. Mater.* **12**, 535-541 (2013), cited in the revised manuscript as Ref. [9] and Ref. [10], respectively). However, these plasmons are not ‘correlated plasmons’ as found in insulating hp-SNO (please see the discussion below).

There is a crucial difference between the plasmons observed in the above mentioned materials (VO₂ and cuprates) and the correlated plasmons observed in our hp-SNO sample. The plasmons observed in VO₂ or cuprates, at least as studied by the above-mentioned groups, *only* appear in their *metallic* or superconducting phases, and disappear when the materials undergo metal-insulator transition into a Mott insulator. This means that the plasmons in these materials are typical conventional plasmons, similar to the conventional plasmons observed in our lp-SNO sample but different from the correlated plasmons observed in the hp-SNO sample. Because the conventional plasmons appear only in the metallic phases of the correlated materials and thus composed of metallic free electrons, this means their behaviour can still be described using Drude model adequately. Indeed, this is the principle behind the use of VO₂ in plasmonic switching and sensing mentioned by Reviewer #1 (Refs. [6-8]): since the plasmon only appear in the metallic phase and disappear in the Mott insulating phase, the plasmon can be switched on and off by heating and cooling the VO₂, which triggers its metal-insulator transition.

On the other hand, the correlated plasmons in hp-SNO appear dominantly in the *Mott-insulating* sample instead of the metallic sample. In fact, the correlated plasmons *diminish* and ultimately *vanish* as the electrical conductivity increases as we explained in our manuscript (paragraph 8 of the Results section). Since in the Mott-insulating sample there should be no free electrons, this means that the correlated plasmons in hp-SNO *cannot* be

explained using simple Drude model, and strong correlation effects should play an important role in the behaviour of these correlated plasmons.

Nevertheless, we do agree that discussing the presence of conventional plasmons in strongly-correlated materials is important, and we have included this in our revised introduction (paragraph 2) accordingly.

- Importantly, with such complex structures such as the perovskite ones, it is known that their electronic and optical properties are also complex and arise from various band splitting, both in the conduction and valence bands. This often occurs right in the region shown by the authors, with no plasmonic effect or correlation effect.

This arises due to spin-orbit coupling, electron-phonon coupling, excitonic effects and anisotropic dielectric functions. For example, see the early work by Mitzi on layered perovskite or the more recent work by the photovoltaics community who are currently assessing the electronic structure of these various perovskite materials (e.g. Even or Fujiwara).

First of all, we would like to remind that one should not make a one-to-one comparison between oxides with halides or other perovskites, because their electronic band structures are different, and therefore their electronic correlations may be different as well.

It has been very established (Refs. [3, 24]) that peaks in the loss function spectra (which are also the spectra measured by electron energy loss experiments) mostly come from plasmonic excitations. Indeed, the three peaks of hp-SNO at 1.7 eV (C_1'), 3 eV (C_2'), and 4 eV (C_3') appear not only in the complex dielectric spectra, but also in the loss function spectra (Fig.

1d). Such plasmonic excitation is supported by our theoretical phenomenological calculations as described in the main article and Supplementary Method.

However, Reviewer #1 argues that the three peaks might instead come from other effects such as spin-orbit coupling, electron-phonon coupling, excitonic effects, and anisotropic effects. Below, we discuss why it is very unlikely that these three peaks of hp-SNO come from these various effects mentioned by Reviewer #1.

1. Spin-orbit coupling

To check the effects of spin-orbit coupling on the band structure and loss function of $\text{SrNbO}_{3.4}$ (which has similar stoichiometry with hp-SNO), we perform DFT calculations with that explicitly include spin-orbit coupling (SOC) effects in its implementation.

To accelerate calculations, $\text{SrNbO}_{3.4}$ is modeled by standard cubic perovskite structure with 5 atoms and primitive cell with 27 atoms respectively. The k-point meshes are $8 \times 6 \times 2$ to sample the Brillouin zone. Number of bands are increased which doubled the default values which up to 256 bands. Symmetries are switched off to compare energies for different orientations and to determine the magnetic anisotropies. Initial magnetic moment for all Nb atoms are parallel to z-direction (out of plane) with $1.0 \mu_B$.

From our calculations, we find that in this material, spin-orbit coupling results in the splitting of the bands with a magnitude of $\sim 0.3\text{-}0.5$ eV. This band splitting is too small compared to the energy separation among the mid-gap peaks, which has a minimum separation of more than ~ 1 eV. (For reference, the separation between C_1 and C_2 peaks is ~ 1.3 eV, while the separation between C_2 and C_3 peaks is ~ 1 eV.)

Furthermore, we also perform random phase approximation (RPA, see the answer to the next question and Ref. [45] for more details) on top of this DFT+ SOC calculation to calculate the loss function (LF) of $\text{SrNbO}_{3.4}$ under the explicit inclusion of spin-orbit coupling effects. We find that the calculated LF spectrum (Fig. R1) only shows one single plasmon peak and is unable to mimic the three mid-gap loss function peaks that we find experimentally in Fig. 1d.

Thus, based on these reasons, we conclude that spin-orbit coupling is not the main reason for the existence of the three mid-gap peaks in hp-SNO.

Figure R1 | Theoretical loss function (LF), $-\text{Im}[\epsilon^{-1}(\omega)]$, of $\text{SrNbO}_{3.4}$ with spin-orbit coupling effects a, The theoretical LF spectra are calculated using density functional theory (DFT) with random phase approximation (RPA) without including strong correlation effects but with an explicit inclusion of spin-orbit coupling implementation.

2. Electron-phonon coupling

Interestingly, phononic peaks due to electron-phonon coupling have actually been observed in $\text{SrNbO}_{3.4}$ (which has similar stoichiometry with hp-SNO), as reported by Ref. [19] that we cited. However, these phononic peaks have very low energies in the range of 5 – 125 meV. Since the intensities of three mid-gap peaks of hp-SNO are very high and their energy positions are in the range of 1.7 – 4 eV, it is very unlikely that these mid-gap peaks come from electron-phonon coupling because their energies are simply too high for phononic interactions in this material.

3. Excitonic effects

First of all, we would like to note that loss function spectrum is not sensitive to excitonic effects. Since the mid-gap peaks of hp-SNO occur in the loss function spectrum, these peaks are not excitonic in nature and instead come from plasmonic excitations.

Furthermore, excitonic peaks in inorganic compounds such as our strontium niobates systems typically have three main characteristics: they are quite sharp (with FWHM of typically less than 200 meV), have asymmetric shape (since exciton interactions can turn into Fano resonances), and have energy positions that are just below the bandgap (especially for Wannier bound excitons, with binding energy, measured from the exciton energy to the first interband transition energy, of typically less than 100 meV). Based on these criteria, it is clear that the three mid-gap peaks of hp-SNO at 1.7 eV, 3 eV, and 4 eV do not originate from excitonic interactions, since they are quite broad (with FWHM of ~500 meV), quite symmetric in shape, and occur quite far below the first interband transition energy of 4.6 eV.

On the other hand, based on these criteria, it is possible for the C_4 peak of hp-SNO (Fig. 1d) to originate from excitonic interactions, since this particular peak is quite sharp, asymmetric, and occur just below the first interband transition, as we have discussed in the manuscript (paragraph 7 of the Results section). As we have further elaborated in the revised manuscript, this C_4 excitonic peak diminishes and ultimately vanishes as the free-charge density increases in conducting mp-SNO and lp-SNO (Fig. 1), because the metallic environment in the conducting films would heavily screen the electron-hole interactions necessary for excitonic excitation. Although at a first glance this behaviour looks similar to the correlated plasmon peaks behaviour (very prominent in insulating hp-SNO but vanishing in conducting lp-SNO due to the heavy screening in the metallic environment), they are fundamentally different because correlated plasmons arise from *electron-electron* interaction while excitons arise from *electron-hole* interaction.

4. Anisotropic effects

We would like to note that we *have* included the possibility of anisotropic properties of the samples in our analysis. As we have detailed in the Method section (paragraphs 3-5), we have used both anisotropic and isotropic models in analysing the ellipsometry data. We find that the lp-SNO and mp-SNO films are slightly uniaxially anisotropic, and the out-of-plane dielectric function of these films are shown in Supplementary Fig. 4. On the other hand, we found that the hp-SNO film, which possesses the three mid-gap peaks, is largely optically isotropic because its in-plane and out-of-plane dielectric functions are the same. From this analysis, we can see that the three mid-gap peaks of hp-SNO are unlikely to come from anisotropic effects, because within our measurement and analysis limits, the hp-SNO film is optically isotropic (see Supplementary Discussion for details).

- The DFT and harmonic coupled oscillator are quite naïve regarding the complexity of the material system, as also mentioned by the other reviewers. As pointed out by the authors, DFT has difficulties capturing the correlation effects of various systems and whether bandgap calculation renders similar experimental results should at least be mentioned in the manuscript.

The fact that our results cannot be explained by DFT calculations, which are not able to properly include correlation effects, further supports that the three mid-gap peaks of hp-SNO do indeed originate from correlation effects. To support this, we perform theoretical calculations of loss function (LF) of SrNbO₃ (which has similar stoichiometry with lp-SNO) and SrNbO_{3.4} (which has similar stoichiometry with hp-SNO) using DFT+PAW with random phase approximation (RPA), and the result is shown in Fig. R2 (and Supplementary Fig. 19 in the revised manuscript). In this calculation, DFT+PAW is used to calculate the electronic band structures of the materials while RPA (Ref. [45]) is used to calculate the LF spectra from these calculated band structures.

Figure R2a shows that according to RPA calculations, the LF of SrNbO₃ should have a conventional plasmon peak and a structure above the first interband gap of 4.6 eV. This is similar to the LF of lp-SNO, albeit with apparent differences in the conventional plasmon peak energy and intensity which can be explained by the tendency of the ground state from DFT calculations to underestimate the bandgap and the density of metallic electrons (Ref. [25]). This ability of DFT and RPA to qualitatively mimic the LF of lp-SNO is not surprising because lp-SNO is a metal with little to no correlation effects.

On the other hand, it can be seen in Fig. R2b that without correlations, DFT and RPA calculations are not able to qualitatively mimic the LF of hp-SNO, particularly the three mid-gap peaks that we observe experimentally in hp-SNO. We note that in RPA, plasmons are treated as superpositions of electron-hole pairs of bare electrons without spectral weight transfers. This time, we cannot attribute the differences due to DFT bandgap underestimation anymore because the experimental LF spectrum contains three plasmonic peaks that cannot be explained by simple Drude model. Based on our analysis, this is where correlations play an important role, and thus it is not surprising that DFT and RPA cannot predict these three peaks because it is unable to properly include correlation effects to begin with. Once again, we note that the DFT calculations that we show in Figs. 6a,6b and Supplementary Fig. 18) are only able to predict the charge confinement effects due to the presence of the extra oxygen planes, but not the correlation resulting from this confinement and much less the mid-gap peaks observed in hp-SNO.

Regarding the bandgap calculation by DFT, we agree that DFT calculations do indeed often underestimate the bandgap of materials (Ref. [25]). This is because DFT can only calculate the ground state of a system, while optical spectra obtained from experiments, including the experimental optical bandgap, come from the excited states. Therefore, when electronic correlations play an important role, DFT-based calculations fail to estimate the experimental optical absorption spectra properly. In fact, how to properly include electronic correlations is one of the main challenges in theoretical studies until now, as shown for example by Ref. [12] that we cited.

We have added this remark about DFT bandgap underestimation in the revised manuscript (paragraph 3 of the Results section). However, once again, it is important to note that the ~ 4.6

eV bandgap that we report in the manuscript is an *experimental* result and not coming from theoretical calculations.

Figure R2 | Experimental loss function (LF), $-\text{Im}[\epsilon^{-1}(\omega)]$, of lp-SNO and hp-SNO films compared with theoretical LF of SrNbO₃ and SrNbO_{3.4}. a, Experimental LF of lp-SNO compared with theoretical LF of SrNbO₃. b, Experimental LF of hp-SNO compared with theoretical LF of SrNbO_{3.4}. The theoretical LF spectra are calculated using density functional theory (DFT) with random phase approximation (RPA) without including strong

correlation effects. We note that in RPA, plasmons are treated as superpositions of electron-hole pairs of bare electrons without spectral weight transfer.

Also, how do changes in the space group affect this analysis with such a small number of atoms in the supercell?

As seen in Figs. 6a,b, Supplementary Fig. 18, and Ref. 17, when the oxygen content of strontium niobates is changed, the spacing between the extra oxygen planes would also change. This in turn would change the symmetry of the unit cell, and consequently change its space group. Thus, the change in space groups between SrNbO_3 , $\text{SrNbO}_{3.3}$, $\text{SrNbO}_{3.4}$, and $\text{SrNbO}_{3.5}$ in our calculations is simply a consequence of their different oxygen contents. As long as we are able to model these changes in symmetry in the calculations properly (particularly since the symmetry changes are only along one direction, *i.e.*, perpendicular to the extra oxygen planes), we should still be able to get qualitatively adequate results even with relatively small numbers of atoms in the supercell. Besides, as we have mentioned in our previous response, our intention with these DFT calculations is only to serve as a qualitative first step in describing these complex materials, especially with the presence of strong correlations in the materials which DFT alone is unable to treat properly and rigorously.

As for the coupled oscillators, I still do not see how correlation connects to these experimental observations.

The advantage of our phenomenological coupled harmonic oscillator model of strongly-correlated plasmons is that we are able to explicitly include the electronic correlations by modelling the Coulomb interaction between neighboring correlated electrons, in the first

approximation, as that of an elastic spring, as shown in Supplementary Fig. 14 (see also the discussions in Supplementary Method). Using this model, we can calculate the optical spectra by controlling correlation effects as manifested in the spring constant, k . For conventional metal, where the electrons are well screened, the k is negligible and our calculations leads to the Drude model of conventional plasmon (see Figs. 6c,f). For correlated system, where the electrons are unscreened, k is not negligible and our calculations are then able to qualitatively mimic the correlated plasmons of thick hp-SNO (see Figs. 6b,g). Below, we further elaborate our thought process when constructing this coupled oscillator model.

First, we would like to reiterate that there is indeed a direct evidence of strong correlations particularly in the hp-SNO sample from the spectral weight analysis as we have explained above. Thus, these strong correlation effects should play an important role in any electronic interactions in the materials, including the collective plasmonic excitations of the electrons.

In non- or weakly-correlated metals, the electron-electron repulsions are mostly screened. Thus, when the plasmonic oscillation of these free electrons is excited, the electrons oscillate as a whole (*i.e.*, in-phase) against the positive ionic background (and not against each other since the electron-electron interactions are largely suppressed). In this case, the restoring force is only due to the Coulomb attraction with the positive ionic background (Fig. 6c and Supplementary Eq. 1), resulting in one bulk plasmon frequency that depends only on the free electron density (which determines the strength of the Coulomb attraction restoring force) and effective mass.

On the other hand, in strongly-correlated materials, electron-electron repulsions are unscreened. Thus, when the plasmonic oscillation of these correlated electrons is excited, the

correlated electrons should also oscillate *against* each other since each electron can now feel the Coulomb repulsions from neighbouring electrons. In this case, the restoring force would also include the electron-electron repulsions, which turns the plasmonic oscillation into a many-body motion with multiple possible natural frequencies (see Supplementary Eq. 10). In the coupled oscillator model, as a first approximation we model the electron-electron interaction with that of an elastic spring. Here, the spring constant represents the strength of the effective electron-electron repulsions and thus the strength of the correlation. As shown in Fig. 6e,g, the strength of our phenomenological model is that even with this semi-classical simple model we are able to qualitatively mimic the experimental data below the bandgap, particularly the three mid-gap peaks of hp-SNO.

Other queries about the paper:

1. The non-systematic work with only three main samples that also have varying thicknesses (> 50 nm difference).

To assure Reviewer 1 about this issue as well as the reproducibility of the correlated plasmons, we prepare a new batch of pressure-dependent samples and measure them using spectroscopic ellipsometry. The new-batch films are prepared more systematically with six different oxygen pressures (as opposed to only three in the main-batch films shown in Fig. 1): 5×10^{-6} (lp-SNO-2), 1×10^{-5} (mlp-SNO-2), 3×10^{-5} (mp-SNO-2), 7×10^{-5} (mhp-SNO-2), 1×10^{-4} (hp-SNO-2), and 5×10^{-4} (hrp-SNO-2) Torr. To minimize thickness-dependent effects, their thicknesses are kept within a narrow range of ~159-182 nm (see Supplementary Table 1). For reference, the thickness of the insulating hp-SNO from the main batch is ~168 nm, so we tried to keep the thicknesses of these new-batch films to be as close to the hp-SNO thickness as possible. Four of the films deposited under lower oxygen pressures (lp-SNO-2, mlp-SNO-

2, mp-SNO-2, and mhp-SNO-2) are conducting, while the two deposited under higher pressures (hp-SNO-2 and hrp-SNO-2) are insulating.

Figure R3 | Complex dielectric function and loss function spectra of new-batch films. a,

Real part of complex dielectric function, $\epsilon_1(\omega)$, of lp-SNO-2, mlp-SNO-2, and mp-SNO-2.

b, Imaginary part of complex dielectric function, $\epsilon_2(\omega)$, and loss function, $-\text{Im}[\epsilon^{-1}(\omega)]$,

spectra of lp-SNO-2. **c,** The $\epsilon_2(\omega)$ and $-\text{Im}[\epsilon^{-1}(\omega)]$ spectra of mlp-SNO-2. **d,** The $\epsilon_2(\omega)$

and $-\text{Im}[\epsilon^{-1}(\omega)]$ spectra of mp-SNO-2. **e,** The $\epsilon_1(\omega)$ spectra of mhp-SNO-2, hp-SNO-2,

and hrp-SNO-2. **f,** The $\epsilon_2(\omega)$ and $-\text{Im}[\epsilon^{-1}(\omega)]$ spectra of mhp-SNO-2. **g,** The $\epsilon_2(\omega)$ and

$-\text{Im}[\epsilon^{-1}(\omega)]$ spectra of hp-SNO-2. **h,** The $\epsilon_2(\omega)$ and $-\text{Im}[\epsilon^{-1}(\omega)]$ spectra of hrp-SNO-2.

Vertical dashed lines indicate the zero-crossings of $\epsilon_1(\omega)$ of lp-SNO-2 (red), mlp-SNO-2 (pink), mp-SNO (purple), and mhp-SNO-2 (light violet). Insets show parts of the spectra zoomed in for clarity.

Figure R4 | Optical conductivity spectra, $\sigma_1(\omega)$, and spectral weight, W , of new-batch films. a, The $\sigma_1(\omega)$ of lp-SNO-2, mlp-SNO-2, mp-SNO-2, mhp-SNO-2, hp-SNO-2, and hrp-

SNO-2. **b**, Evolution of W of three energy regions: W_1 (0.6-2.0 eV), W_2 (2.0-4.3 eV), and W_3 (4.3-6.3 eV) across the three films. The W_{tot} is the total W from 0.6 to 6.3 eV.

From Fig. R3 and Supplementary Fig. 14, we can see that the behaviours of the correlated and conventional plasmons as well as excitonic peaks in this new batch are very similar to what observed in the main-batch films shown in Fig. 1. Furthermore, Fig. R4 and Supplementary Fig. 16 also show that the pressure-dependent metal-insulator transition is reproducible as well, and that the films deposited under higher oxygen pressures consistently have the signature of strong electronic correlations. This indicates that the correlated plasmons are indeed reproducible (and not only appear in one particular sample), and its pressure-dependent behaviour can be reproduced consistently while keeping their film thicknesses almost constant. Below (and in the revised Supplementary Discussion), we discuss this issue in more details.

From Fig. R3g and Supplementary Fig. 14g, it can be seen that the $\epsilon_2(\omega)$ and LF, spectra of the insulating hp-SNO-2 film (deposited under the same high oxygen pressure as the main-batch hp-SNO) also show the distinct signatures of multiple correlated plasmon peaks. Interestingly, the number of correlated plasmon peaks in hp-SNO-2 (five) is higher than in hp-SNO (three), with two new correlated plasmon peaks appearing in the lower energy region below ~ 1.4 eV that were absent in hp-SNO. Furthermore, the energies of the three highest correlated plasmon peaks in hp-SNO-2 (~ 2.4 , ~ 3.5 , and ~ 4.1 eV) are slightly higher than the three correlated plasmon peaks of hp-SNO (~ 1.7 , ~ 3.0 , and ~ 4.0 eV). These indicate that the new-batch hp-SNO-2 film might have a slightly higher electronic correlation than hp-SNO, as corroborated by its resistivity that is ~ 11 times higher than that of hp-SNO (see

Supplementary Table 1). Note that at this stage it is still challenging to fully control dynamical processes that can affect film growth in pulsed-laser depositions, which might cause the slight differences between the sample batches. Nevertheless, our main conclusions do not depend on it, especially since the extra correlated plasmon peaks can be well explained within the framework of our coupled oscillator model and the general behaviour of the correlated plasmons in this new batch is very similar to that in the main batch, as further discussed below.

From Supplementary Fig. 21, it can be seen that when the number of oscillators between the oxygen wall is increased above seven, more correlated plasmon peaks start to appear especially at the lower energy region. Thus, the appearance of extra correlated plasmon peaks in hp-SNO-2 (and hrp-SNO-2, see below) can be explained by these higher-order modes in our coupled oscillator model. More importantly, the appearance of distinct correlated plasmon peaks in hp-SNO-2 indicates that the correlated plasmons are reproducible and not only appear in one particular sample.

In higher-pressure hrp-SNO-2, the correlated plasmon peaks are even more prominent (particularly the two lowest-energy peaks) and slightly blue-shifted. On the other hand, when free-charge density is increased in mhp-SNO-2 and the film becomes conducting, these correlated plasmons start to diminish (only two correlated plasmons at ~ 2.3 and ~ 3.9 eV are observed in mhp-SNO-2) and ultimately vanish in lp-SNO-2, consistent with correlated plasmon behaviour of the main-batch films shown in Fig. 1.

The behaviors of conventional plasmon and excitonic peaks in these new-batch films are also very similar with those of the main-batch films. From Fig. R3b and Supplementary Fig. 14b, it can be seen shows that lp-SNO-2 film has the highest conventional bulk plasmon energy (among the new-batch films) at ~ 1.8 eV. This conventional plasmon gradually red-shifts as the free-charge density decreases (down to ~ 1.2 eV in mhp-SNO-2), and ultimately vanishes in insulating hp-SNO-2 and hrp-SNO-2 films. This is consistent with the conventional plasmon behaviour of the main-batch films in Fig. 1 and in contrast with the correlated plasmon behaviour described above. Furthermore, the insulating hp-SNO-2 and hrp-SNO-2 films also show a distinct excitonic peak at ~ 4.5 eV (Figs. R3g,h and Supplementary Figs. 14g,h), and this excitonic peak gradually diminishes and ultimately vanishes as free-charge density increases, similar to the excitonic peak behaviour of the main-batch films (Fig. 1).

Optical conductivity and W analyses of these new-batch films (Fig. R4 and Supplementary Fig. 16) also show that their spectral weight transfer behaviour is very similar to that of the main-batch films. As W_1 (Drude region) decreases from lp-SNO-2 to mhp-SNO-2, W_3 (first interband transition region) increases due to the increased availability of unoccupied Nb-4d states. However, when MIT happens between mhp-SNO-2 and hp-SNO-2, *both* W_1 and W_3 *anomalously decrease*, leading to an overall decrease of W below 6.3 eV (W_{tot}). Again, according to the f -sum rule (Refs. [24, 29, 30]), this decrease has to be compensated by an equivalent increase above 6.3 eV, implying anomalous wide-range spectral weight transfers on the onset of the MIT, which, we repeat, is a direct evidence of strong correlations in hp-SNO-2 (Refs. [11, 24, 30-35]). The W_3 and W_{tot} continue to decrease in hrp-SNO-2, implying even larger anomalous spectral weight transfers and thus stronger electronic correlations than hp-SNO-2, which enhances the correlated plasmons as shown in Fig. R3h and Supplementary Fig. 14h. Thus, it can be seen that the pressure-dependent MIT is also reproducible, and that

the films deposited under higher oxygen pressures are consistently shown to have the signature of strong electronic correlations.

We have added this discussion in paragraph 15 of the Results section of the revised main article, Supplementary Discussion paragraphs 4-7 of the revised supplementary, and the supporting figures have been added as Supplementary Figs. 12-17.

The only thickness-dependent measurement carried out does not support their arguments of correlations. Instead, it supports the fact that there are various perovskite structures with varying optical/excitonic transitions.

Even though the thickness-dependent set of samples have different thicknesses, they are still made of the same compound, which means that we should be able to compare them against each other and find the reasons and mechanisms of why their properties can be different compared to each other.

In this case, we find that these thickness-dependent high-pressure samples can be compared in terms of their spectral weight transfers, which, as we have discussed above, is a very established and direct way to gauge the correlations in a set of materials. From this analysis, we come to the conclusion that their different optical excitation signatures are caused by the difference in their correlation strength, as we have discussed extensively in the manuscript (paragraphs 9 and 10 of the Results section of the revised manuscript). As the high-pressure film becomes thinner, the overall spectral weight below our measurement limit of 6.5 eV steadily increases (see Fig. 5). Since spectral weight obey the f -sum rule, which is a charge

conservation rule, this means that as the films become thinner, the spectral weight transfers into energy *above* 6.5 eV have to, conversely, steadily decrease.

Previously, we have discussed that, compared to the metallic low-pressure samples, the decrease in spectral weight below 6.5 eV in insulating, thick high-pressure sample (hp-SNO) is ought to be caused by anomalous spectral weight transfer into energies above 6.5 eV, which is a wide energy range for a spectral weight transfer. According to very established studies (Refs. [11, 24, 30-35]), this is a direct evidence of strong correlations in the thick hp-SNO sample. Thus, since the anomalous spectral weight transfer decreases as the high-pressure film becomes thinner, this means that the correlation strength in these thinner films also decreases as well.

Lastly, we would like to again state that the mid-gap peaks found in these thinner high-pressure films are indeed plasmonic in origin or at least have a strong coupling with plasmonic effects, and not coming from ordinary optical (*i.e.*, band-to-band) or excitonic transitions, for reasons similar to the mid-gap peaks found in the thicker high-pressure film (hp-SNO). The main reason is that the mid-gap peaks (at ~3.2 eV and ~4.4 eV for 81 nm film; at ~4.0 eV for 52 nm film) occur in the loss function spectra, which is a strong indication that these peaks are plasmonic in origin or at least have a strong coupling with plasmonic effects. The mid-gap peaks are unlikely to come from ordinary band-to-band transitions because non-correlated DFT calculations are unable to predict the existence of these peaks even after incorporating spin-orbit coupling effects (see Fig. R1), and they are not excitonic peaks because they are too broad, largely symmetric in shapes, and most occur at energies too far below the bandgap energy of ~4.6 eV.

Also, other peaks - at 2.25 (between C1 and C2)

As this peak (which, upon closer look, actually appears at ~ 2.4 eV instead of 2.25 eV) also occurs in the loss function spectra (Fig. 1d), it should also be plasmonic in origin, like C₁, C₂, and C₃ peaks. From Supplementary Fig. 20, we can see that when we model the correlated plasmons with the coupled oscillator model, we can adjust the number of oscillator in one chain. To qualitatively mimic the C₁, C₂, and C₃ peaks in both the dielectric function and loss function spectra, it turns out we need to have 7 oscillators in one chain to have the closest fit (paragraph 6 of the Discussion section and Supplementary Method). However, other numbers of oscillators are also possible, and as seen in Supplementary Fig. 21 higher numbers of oscillators tend to result in higher numbers of correlated plasmon peaks as well. Thus, we determine that the peak at 2.4 eV should come from these higher-order modes. Nevertheless, this 2.4 eV peak is quite small compared to C₁, C₂, and C₃ peaks, which means that the 7-coupled oscillator mode should still be the dominant mode of oscillation.

and 6 eV (after C₅) - are not addressed in the paper. There is also a peak between B₃ and B₄, albeit small potentially, that is not mentioned in the paper.

We apologize for this confusion. The peak centred at ~ 6 eV comes from the same transition as the 4.6 eV peak, which is the interband O-2*p* to Nb-4*d* transition, albeit happening at different points in the momentum space. This is the reason why we initially group the two transitions at 4.6 eV (which extends up to 5.5 eV) and 6 eV under one name (C₅). The valence band (in this case, O-2*p*) and conduction band (Nb-4*d*) can have different dispersions in the momentum space. Thus, at different points in the momentum space, the energy gap between these two bands can vary. The 4.6 eV peak happens at the point where the two bands

are the closest in energy gap (*i.e.*, its direct bandgap), while the 6 eV transition peak happens at the point where the two bands have a 6 eV energy gap. To avoid further confusion, in the revised manuscript we have properly labelled the 4.6 eV peak (which extends up to 5.5 eV) as the C₅ peak and the 6 eV peak as the new C₆ peak.

The same reason also applies for the small peak at 4.6 eV (which extends up to 5.2 eV) before B₄ pointed out by Reviewer #1. This small peak also has the same origin with the broader peak centred at ~5.6 eV, which is the interband transition of O-2*p* to Nb-4*d*, but happening at different points in the momentum space. Again, to avoid further confusion we have labelled this small peak as the new B₄ peak, while the broader peak centred at 5.6 eV is labelled as the new B₅ peak.

Furthermore, for consistency we have also split the A₂ peak of lp-SNO in the revised manuscript into two peaks: one at ~4.6 eV to ~5.3 eV (relabelled as the new A₂ peak) and another at energies above 5.3 eV (relabelled as the new A₃ peak). Note that in this sample (as well as in mp-SNO) the two peaks have become too broadened and it is difficult to resolve them as two distinct peaks. The same as hp-SNO and mp-SNO, these two peaks come from the same interband transition of O-2*p* to Nb-4*d*, but happening at different points in the momentum space.

2. The author discusses the advantages of surface plasmon in the introduction as motivation, but later on proceed to discuss that this is only a bulk effect, with these peaks vanishing quite rapidly as the sample thickness is decreased.

Since the audience of *Nature Communications* are from a wide range of disciplines, readers might not be immediately familiar with plasmonics and its advantages. Thus, in the introduction we need to give the audience a general idea of the usefulness and potential applications of plasmonics. Currently, surface plasmons are the most active and advanced research area in plasmonics, which is why we discuss about it the most in the introduction as the motivation for our research in plasmonics in general. Unfortunately, the research in bulk plasmons is not as active, at least to our knowledge. With our discovery of correlated plasmons in this paper, we hope to be able to open a new path for plasmonics research, even as a bulk effect. Further research in these correlated plasmons may even find its surface equivalent, which can be more directly compared and integrated with conventional surface plasmons, but this is beyond our current scope.

3. What is the lattice mismatch between the various structures and the LCMO substrate?

We thank Reviewer #1 for bringing the issue up.

First, we would like to clarify that the substrate that we use in the study is (001) LaAlO_3 , not LCMO, which, we assume, is short for $(\text{La,Ca})\text{MnO}_3$.

The lattice constant of LaAlO_3 is 3.791 \AA (Ref. [36]), while the in-plane lattice constant of our strontium niobate films is 4.04 \AA (Supplementary Fig. 1), resulting in a lattice mismatch of $\sim 6.6\%$, which is relatively large. This relatively large mismatch might be one of the reasons behind the thickness-dependent behaviours of the high-pressure films (Figs. 4,5). In thinner films, this relatively big lattice mismatch at the film-substrate interface should play a more significant role in influencing the electronic band structure of the whole film. This large

lattice mismatch can be one of the reasons why thinner films have less correlated plasmons excitations (and ultimately no correlated plasmons in the 20 nm film) compared to thicker films. The interface effects due to this large lattice mismatch may suppress the electron-electron correlations and thus the excitations of correlated plasmons in the thinner films. On the other hand, in thicker films the lattice mismatch should play a smaller role because their thickness should allow them to have more complete relaxations, which might be the reason why thicker films have stronger correlations and thus more correlated plasmon excitations compared to thinner films.

We have added this into the discussion about the thickness-dependent behaviour of the high-pressure films (paragraph 14 of the Results section).

4. From Fig3, it appears there are mixed states present (sample inhomogeneity) which result also in this anisotropic dielectric functions. How were these two samples (3b and 3c) modeled differently?

First, we would like to clarify that the one shown in Fig. 3b is the mid-pressure mp-SNO film, while the one shown in Fig. 3c is the high-pressure hp-SNO film. The mp-SNO film is a mixture or intermediate phase between hp-SNO and the low-pressure lp-SNO film, since its atomic microstructure, oxygen content, electrical properties, and complex dielectric function are in between these two other films.

From Supplementary Fig. 8, we have shown that quantitatively, the number of extra oxygen planes in mp-SNO is four times lower than in hp-SNO. Thus, even though at a first glance their microstructures might look similar in Figs. 3b and 3c, in terms of stoichiometry they are

actually quite different. In the introduction (paragraph 3), we have discussed that in this particular family of oxides, even small changes in oxygen stoichiometry can lead to different electronic properties. Thus, it is not surprising that mp-SNO and hp-SNO can have different electronic properties and thus need to be modelled differently.

In the spectroscopic ellipsometry analysis (see paragraphs 3-5 of the Method section), we have modeled all samples with both anisotropic and isotropic models. After the analysis, we found that only the lp-SNO (which has no extra oxygen planes) and mp-SNO (which only has small number of extra oxygen planes present, at least compared to hp-SNO) films are uniaxially anisotropic and thus need to be treated as such. On the other hand, the hp-SNO film (which has the highest number of extra oxygen planes) is largely optically isotropic instead, since its in-plane (ordinary) and out-of-plane (extraordinary) complex dielectric function are the same.

Reviewer #2 (Remarks to the Author):

Since the authors have adequately addressed my comments made in the first run of review, I am ready to accept the revised version for publication in Nature Communications after the authors have made the following minor revision.

- Line 21, Page 2: note that localized surface plasmons have also been observed in the metallic state of VO₂ - a phase changing material and used to probe the role of defects in the phase transition of VO₂ nanoparticles, see Nano Letters 2012, 12, 780-786. The authors should cite this representative work in order to make the literature review more complete.

We would like to thank Reviewer #2 for his/her support for our manuscript. We have included the reference suggested by Reviewer #2 as the new Ref. 6 and discussed it in the introduction (paragraph 2) of our revised manuscript.

Reviewer #3 (Remarks to the Author):

I am satisfied by the changes in the manuscript made by the authors and have found their reply to the referee's criticism adequate and convincing. I think the paper may be published in the present form.

We would like to thank Reviewer #3 for his/her support for our manuscript.

Reviewer's comments:

Reviewer #1 (Remarks to the Author):

Please accept the manuscript since they have addressed all lingering concerns.

Response:

We thank Reviewer #1 for his/her support for our paper.